# Slit homogenizer introduced performance gain analysis based on Sentinel-5/UVNS spectrometer

Timon Hummel[1,2], Christian Meister[2], Corneli Keim[3], Jasper Krauser[2], and Mark Wenig[1]

[1]Meterological Institute LMU Munich, Theresienstraße 37, Munich, Germany
[2]Airbus Defence and Space GmbH, Willy-Messerschmitt-Str. 1, 82024 Taufkirchen, Germany
[3]Airbus Defence and Space GmbH, Claude-Dornier-Str., 88090 Immenstaad, Germany

**Correspondence:** Timon Hummel (timon.hummel@airbus.com)

**Abstract.** Spatially heterogeneous Earth radiance scenes affect the atmospheric composition measurements of high resolution Earth observation spectrometer missions. The scene heterogeneity creates a pseudo-random deformation of the instrument spectral response function (ISRF). The ISRF is the direct link between the forward radiative transfer model, used to retrieve the atmospheric state, and the spectra measured by the instrument. Hence, distortions of the ISRF owing to radiometric inhomogeneity of the imaged Earth scene will degrade the precision of the Level-2 retrievals. Therefore, the spectral requirements of an instrument are often parametrized in the knowledge of the ISRF over non-uniform scenes in terms of shape, centroid position of the spectral channel and the Full Width at Half Maximum (FWHM).

The Sentinel-5/UVNS instrument is the first push-broom spectrometer that makes use of a concept referred as slit homogenizer (SH) for the mitigation of spatially non-uniform scenes. This is done by employing a spectrometer slit formed by two parallel mirrors, scrambling the scene in along track direction (ALT) and hence averaging the scene contrast only in the spectral direction. The flat mirrors do not affect imaging in the across track direction (ACT) and thus preserve the spatial information in that direction. The multiple reflections inside the SH act as coherent virtual light sources and the resulting interference pattern at the SH exit plane can be described by simulations using scalar diffraction theory.

By homogenizing the slit illumination, the SH strongly modifies the spectrograph pupil illumination as a function of the input scene. In this work we investigate the impact and strength of the variations of the spectrograph pupil illumination for different scene cases and quantify the impact on the ISRF stability for different types of aberrations present in the spectrograph optics.

## 1 Introduction

The Ozone Monitoring Instrument (OMI) was the first instrument identifying the issue arising from non-uniform Earth scenes on the shape and maximum position of the spectral response of the instrument (Voors et al., 2006). In grating based imaging spectrometers, the Earth ground scene is imaged by the telescope onto the instrument entrance slit plane. The scanning over the ground area is achieved by either a scanning mirror or a push-broom configuration, where different areas of the surface are imaged as the satellite flies forward. In the subsequent spectrograph, the slit illumination gets spectrally resolved by a dispersive element and re-imaged on the focal plane array (FPA) by an imaging system. The limited spectral resolving power of the instrument arising from diffraction and aberration is described by a convolution of the slit image with the spectrometer and

25 detector point spread functions (PSF). In this study, we interpret the resulting intensity pattern on the FPA in spectral direction as the Instrument spectral response function (ISRF). In fact, there exist other definitions of the ISRF. The differentiation of the definitions become particularly important in the presence of spectrometer smile effects (Caron et al., 2017). As we neglect such effects, we will continue with the previously described definition of the ISRF.

Depending on the observed scene heterogeneity, the entrance slit will be inhomogeneously illuminated. In the case of a classical
slit, this will alter the shape of the ISRF (see Fig. 1). Moreover, a scene dependency in the PSF will also affect the ISRF, which will be particularly discussed in this manuscript. As the ISRF is the direct link between the radiative transfer model and the spectrum measured by the instrument, a scene dependent shape of the ISRF will have an immediate impact on the accuracy of the Level-2 retrieval products. Figure 2 depicts a representative Top-of-Atmosphere spectrum (SZA 10°, albedo 0.05) for the Sentinel-5/UVNS (Ultra-Violet/Visible/Near-Infrared/SWIR) SWIR-3 spectrometer, incident on the instrument's entrance
aperture. The monochromatic spectrum will be smeared by means of a convolution with an exemplary ISRF, which depends on the imaging properties of the instrument for any given wavelength. In general, the ISRF is a wavelength and field-of-view dependent instrument characteristic and hence varies over the FPA position. It is experimentally determined prior to launch in on-ground characterization campaigns. Whenever the in-orbit ISRF shape deviates from the on-ground characterized shape, due to for example heterogeneous scenes, it will affect the measured spectrum, from which the Level-2 products are retrieved
(e.g. $CH_4$ and $CO$ in the SWIR-3 channel of Sentinel-5/UVNS).

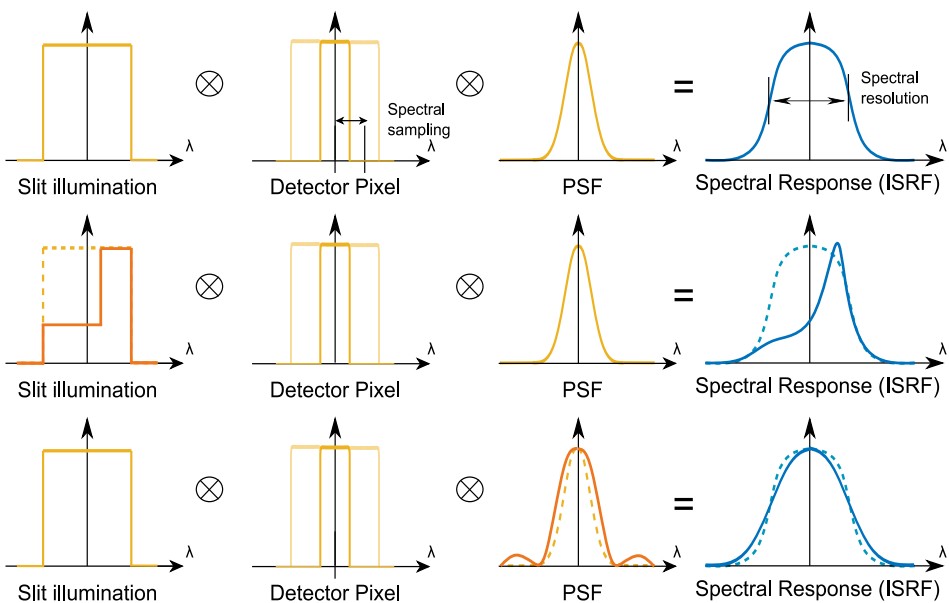

**Figure 1.** The ISRF of an imaging spectrometer is given by the convolution of the slit illumination, pixel response and the optical PSF of the spectrograph optics. In the context of heterogeneous scenes, the ISRF can be altered due to non-uniform illumination and instabilities in the optical PSF. This leads to deformation in the ISRF with respect to the centroid, shape and the FWHM.

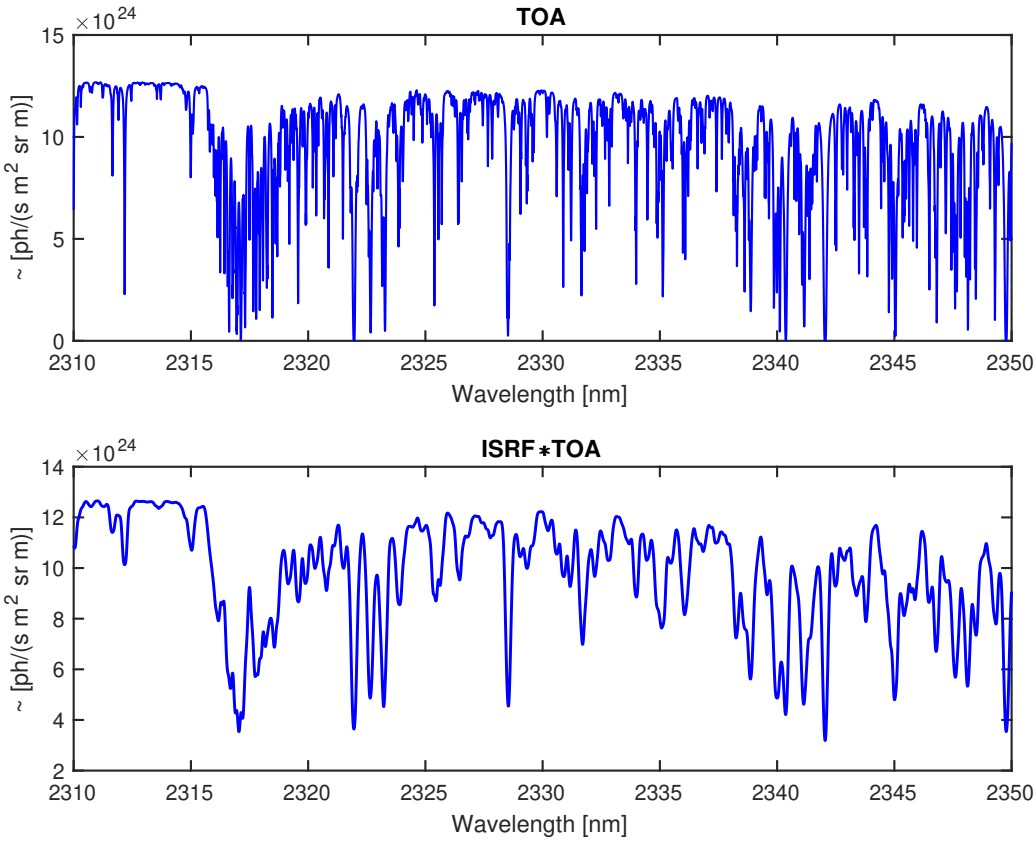

**Figure 2.** (Top) Representative high-resolution Earth Top-of-Atmosphere (TOA) spectrum incident on a space-borne instrument. The structures originate from the absorption features by $CH_4$, CO and $H_2O$. (Bottom) TOA spectrum convolved with a constant exemplary ISRF. Whenever the ISRF deviates from the the on-ground characterized shape, the measured spectrum, which sets the basis for the retrieval algorithms, will be altered.

This effect is particularly prominent for instruments with a high spatial resolution. The along track motion of the satellite during the integration times results in a temporal averaging of the ISRF variation, which reduces the impact of scene heterogeneity. The impact of e.g. albedo variations depends on the instantaneous field-of-view (IFOV) and the sampling distance in ALT (for Sentinel-5/UVNS: FoV = 2.5 km, ALT SSD = 7 km). Spectrometers with a large scan area like GOME (Burrows et al., 1999) or SCIAMACHY (Bovensmann et al. (1999), Burrows et al. (1995)) are less vulnerable to contrast in the Earth scene due to the small ratio between the slit footprint and the smear distance. In contrast, recent high resolution hyperspectral imaging spectrometer with IFOV comparable to the sampling distance (or scan area) are more strongly affected and therefore demand a set of stringent requirements on the inflight knowledge and stability of the ISRF. This is necessary, as distortions in the ISRF due to non-uniform scenes will introduce biases and pseudo-random noise in the Level-2 data and therefore in the precision of atmospheric composition products. For Sentinel-5 Precursor (S5P) satellite (Veefkind et al., 2012), launched in

2017, with the Tropospheric Monitoring Instrument (TROPOMI) being the single payload, Hu et al. (2016) showed that the stability and knowledge of the ISRF is the main driver of all instrument calibration errors for the retrieval accuracy. Landgraf et al. (2016) estimate the error of the retrieved CO data product due to non-uniform slit illumination to be in the order of 2 % with a quasi random characteristics. Noël et al. (2012) quantify the retrieval error for the upcoming Sentinel-4 UVN imaging spectrometer for the tropospheric $O_3$, $NO_2$, $SO_2$ and HCHO. They identify a difference in the retrieval error depending on the trace gas under observation. The largest error occurs for $NO_2$ with a mean error of 5 % and a maximum error of 50 %. They propose a software correction algorithm, which is based on a wavelength calibration scheme individually applied to all Earth radiance spectra. As discussed by Caron et al. (2019), this type of software correction can only be applied to dedicated bands (UV,VIS,NIR) but is failing particularly in the SWIR bands due to the strong absorption lines of highly variable atmospheric components.

Sentinel-5/UVNS (Irizar et al., 2019) is the first push-broom spectrometer that employs an onboard concept to mitigate the effect of non-uniform scenes in the along-track direction. A hardware solution called slit homogenizer (SH) is implemented which reduces the scene contrast of the Earth radiance in the along track direction (ALT) of the satellite flight motion by replacing the classical slit with a pair of two parallel extended mirrors (Fig. 3a). The two parallel rectangular mirrors composing the entrance slit have a distance of $b = 248\,\mu m$ , side lengths of $65\,mm$ in ACT and a length of $9.91\,mm$ (SWIR-3) along the optical axis. Thereby, the light focussed by the telescope optics onto the slit entrance plane is scrambled by multiple reflections in the ALT direction, whereas in ACT the light passes the SH without any reflection. Heterogeneous scenes in ACT direction may also affect the ISRF stability in the presence of spectrometer smile. This effect will not be covered in this study and instead we refer the reader to Gerilowski et al. (2011) and Caron et al. (2017). For a realistic reference Earth scene of the Sentinel-5/UVNS mission provided by ESA (Fig. 5), the ISRF shall meet the requirements of $< 2$ % ISRF shape knowledge error, $< 1$ % relative Full width half Maximum (FWHM) knowledge error and $0.0125\,nm$ centroid error in the SWIR-3. Meister et al. (2017) and Caron et al. (2019) presented simulation results providing a first order prediction of the performance of the SH principle, which are relevant to achieve the performance requirements above. However, so far several second order effects haven't been quantitatively addressed in the prediction of the homogenizing performance. This paper extents the existing first-order models and provides a more elaborated and comprehensive description of the SH and its impact on performance and instrument layout. We present an end-to-end model of the Sentinel-5/UVNS SWIR-3 channel ($2312\,nm$). In particular, we determine the spectrograph pupil illumination which is altered by the multiple reflections inside the SH. This effect changes the weighting of the aberrations present in the spectrograph optics and consequently results in a scene dependency in the optical PSF. As the ISRF is not only a function of the slit illumination, but also of the spectrograph PSF, a variation in the intensity distribution across the spectrograph pupil will ultimately put an uncertainty and error contribution to the ISRF. The severity of the spectrograph illumination distortion highly depends on the slit input illumination and the strength and type of aberrations present in the spectrograph. In order to quantify the achievable ISRF stability, we simulate several input scenes and different type of aberrations.

The outline of this paper is as follows: Sect. 2 describes the model we deployed to propagate the light through the SH by Huygens-Fresnel-diffraction formula. Applying Fourier optics, we formulate the propagation of the complex electric field

from the SH exit plane up to the grating position, representing the reference plane for the evaluation of the spectrograph pupil intensity distribution. In Sect. 3 we quantify the spectrograph pupil intensity distribution for several Earth scene cases. The scene dependent weighting of the aberrations in the spectrograph and its impact on the ISRF properties is discussed and quantified in Sect. 4. Finally, we summarize our results in Sect. 5.

## 2 Slit Homogenizer Model

This section describes the underlying models and the working principle of the SH. The first part briefly summarizes the model developed by Meister et al. (2017), which propagates the field through the Sentinel-5/UVNS instrument up to the SH exit plane by using a scalar-diffraction approach. In the second part a novel modelling technique of the spectrograph optics is introduced. We put a particular focus on the scene dependency of the spectrograph illumination while using a SH.

### 2.1 Near-Field

The light from objects on the Earth, that are imaged at one spatial position (along slit) within the homogenizer entrance slit, arrive at the Sentinel-5/UVNS telescope entrance pupil as plane waves, where the incidence angle $\theta$ is between $\pm 0.1°$. The extent of the wavefront is limited by the size and shape of the telescope aperture. Neglecting geometrical optical aberrations, the telescope would create a diffraction limited point spread function in the telescope image where the SH entrance plane is positioned. Depending on the angle of incidence, the PSF centroid will be located at a dedicated position within the SH entrance plane. The electric field of the diffraction pattern in the SH entrance plane is given as the Fourier transform of the complex electric field over the telescope pupil. For a square entrance pupil, the diffraction pattern is calculated as: (Goodman, 2005, p.103)

$$\tilde{U}_{f,\theta}(u_a, v_a) = \frac{A}{i\lambda f} e^{i\frac{k}{2f}(u_a^2 + v_a^2)} \int_{\Omega} e^{iky_t\,sin(\theta)} e^{-i\frac{k}{f}(x_t u_a + y_t v_a)}\, du_a dv_a \tag{1}$$

$$= \frac{iAD^2}{\lambda f} e^{i\frac{k}{2f}(u_a^2 + v_a^2)} sinc\left(\frac{Dk}{2f}u_a\right)\, sinc\left(\frac{Dk}{2f}\left(f sin(\theta) - v_a\right)\right) \tag{2}$$

where $(x_t, y_t)$ are the coordinate positions in the telescope entrance pupil and $(u_a, v_a)$ are the respective coordinates in the SH entrance plane. $\Omega$ denotes the two-dimensional entrance pupil area, $f$ is the focal length of the telescope, $A$ the amplitude of the plane wavefront at the telescope entrance pupil, $D$ the full side length of the quadratic telescope entrance pupil and $k = \frac{2\pi}{\lambda}$ the wavenumber. Further, the relation $\int_{-a}^{a} e^{ixc} = 2a sinc(ca)$ and a Fresnel approximation was applied in Eq. (2). The propagation of $\tilde{U}_f$ through the subsequent SH is described by the Huygens-Fresnel principle (Goodman, 2005, p. 66). The reflections at the two mirrors are accounted for by inverting the propagation component in ALT upon every reflection $n$ as

$$U_{f,\theta}(u_a, v_a) = R^{|n|} e^{in\pi} \tilde{U}_{f,\theta}(u_a, (-1)^n(v_a - nb)), \qquad \text{for } v_a \in \left[-\frac{b}{2} + nb, \frac{b}{2} + nb\right] \tag{3}$$

where $R$ is the reflectivity, $b$ is the slit width and $e^{in\pi}$ describes a phase jump upon every reflection $n$. Inserting Eq. (2) into (3) and applying the Huygens-Fresnel diffraction principle yields the expression for the intensity distribution at the SH exit plane

for a given incidence angle $\theta$, SH length $l$ and position $r(u_a, v_a) = \sqrt{l^2 + (u_b - u_a)^2 + (v_b - v_a)^2}$ as

$$U_\theta(u_b, v_b) = \frac{lAD^2}{\lambda^2 f} \int\limits_{u_a \in \mathbb{R}} \int\limits_{v_a = -\frac{b}{2}}^{v_a = \frac{b}{2}} \sum_{n \in \mathbb{N}} R^{|n|} \frac{e^{i\frac{k}{2f}\left(u_a^2 + ((-1)^n (v_a - nb))^2\right) + ikr(u_a, v_a + nb) + in\pi}}{r^2(u_a, v_a + nb)}$$

$$\cdot \, sinc\left(\frac{Dk}{2f} u_a\right) sinc\left(\frac{Dk}{2f}\left(f\,sin(\theta) - (-1)^n\, v_a\right)\right) du_a dv_a \tag{4}$$

where $u_b, v_b$ are the coordinates of the position at the SH exit plane. Evaluating Eq. (4) for every incidence angle of the Sentinel-5/UVNS field of view (FoV) results in the so called SH transfer function (Fig. 3b ), which maps any field point originating from Earth to an intensity distribution at the SH exit plane. In a purely geometric theory and a perfect SH configuration in

terms of length, every point source would be distributed homogeneously in ALT direction (Fig. 3a). However, as is quantified in Eq. (4), the field distribution at the SH output plane highly depends on interference effects due to path differences of the reflected light inside the SH, resulting in a non-uniform transfer function as shown in Fig. 3b.

A full experimental validation of the propagation model through the SH is still missing. An initial approach to validate the model in a breadboard activity was conducted by ITO Stuttgart and published in Irizar et al. (2019).

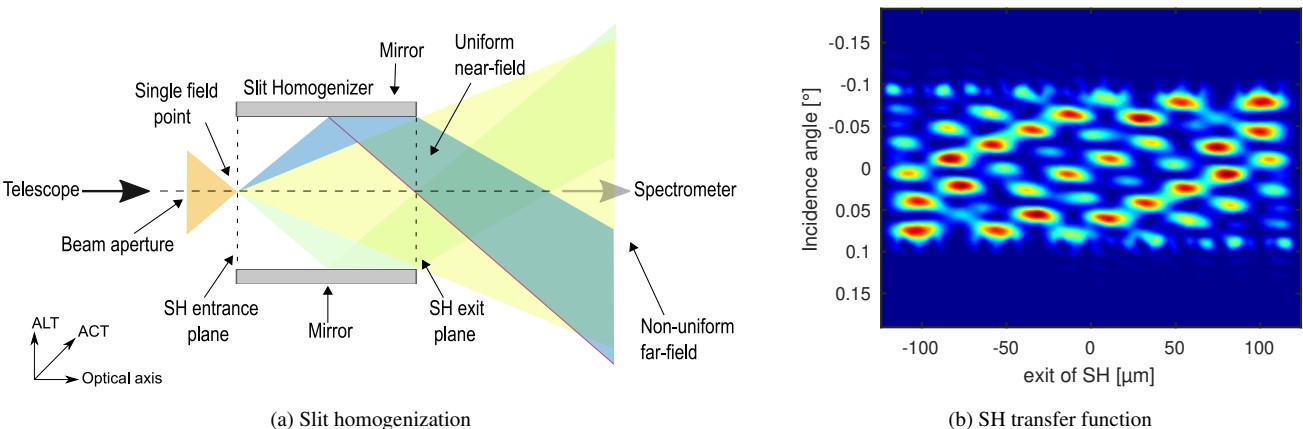

(a) Slit homogenization                           (b) SH transfer function

**Figure 3.** (a) The SH homogenization principle based on a purely geometrical concept. With an appropriate length selection, the SH would perfectly homogenize any input scene. (b) SH transfer function. In reality, the output pattern of the SH is strongly affected by interference effects, resulting in a complex illumination pattern at the slit exit.

**2.2   Far-Field**

In a space-based imaging spectrometer equipped with a classical slit acting as a field stop, a point source on the Earth surface enters the instrument as a plane wavefront with a uniform intensity over the telescope pupil. As this principle applies for every point source in a spatial sample on the Earth, the telescope pupil intensity homogeneity is independent of the radiance variation among the point sources in a spatial sample. Besides some diffraction edge effects in the slit plane, the telescope

pupil intensity distribution gets retrieved in the spectrograph pupil. This is not the case when introducing a mirror based SH.

Existing SH models (Meister et al. (2017) and Caron et al. (2019)) implement the spectrometer as a simple scaling factor and the ISRF on the FPA is obtained via the convolution of the SH output intensity distribution, the pixel response implemented as a characteristic function and the spectrograph PSF. In this contribution we model the propagation through the spectrograph more accurately by including the spectrograph optics, such as the collimator, a dispersive element and the imaging optics. In particular, the inclusion of these optical parts becomes important because the SH not only homogenizes the scene contrast in the slit, but it also significantly modifies the spectrograph pupil illumination. A schematic diagram of the SH behaviour and the instrument setup is shown in Fig. 4. A plane wavefront with incidence angle $\Theta$ is focussed by a telescope on the SH entrance plane. In ACT direction, the light is not affected by the SH. After a distance $l$, corresponding to the SH length, the diffraction limited PSF at the SH entrance plane is converted to the far-field pattern of the diffraction pattern. Independent of the applied scene in ACT, the telescope pupil intensity distribution in ACT is mostly retrieved again at the spectrograph pupil. The exact distribution of the spectrograph pupil illumination is affected by magnification factor and a truncation of the electric field at the SH entrance plane, which leads to a slight broadening and small intensity variations with a high frequency in angular space (Berlich and Harnisch, 2017). In ALT the diffraction pattern in the SH entrance plane undergoes multiple reflections on the mirrors, so that eventually the whole exit plane of the SH is illuminated. To preserve the full image information along the swath, the entrance plane of the SH must be imaged; to homogenize the scene in ALT the exit plane of the SH must be imaged. This is achieved by an astigmatism in the collimator optics. Moreover, the multiple reflections inside the SH lead to a modification of the system exit pupil illumination. In other words, the SH output plane (near-field) and the spectrograph pupil intensity variation (far-field) strongly depend on the initial position of the incoming plane wave, and therefore on the Earth scene radiance in ALT direction. Following a first simple geometrical argument as discussed by (Caron et al., 2019), we consider a point source at the SH entrance. The rays inside the cone emerging from this source will undergo a number of reflections depending on the position of the point source and the angle of the specific ray inside the cone. The maximum angle is given by the telescope F-Number. With this geometrical reasoning it becomes obvious, that the number of reflections differs among the rays inside the cone. If the number of reflections is even, a ray keeps its nominal pupil position; whereas if the number is odd, its pupil coordinate will be inverted. From this argument we deduce that the spectrograph pupil illumination will be altered with respect to the telescope pupil illumination. Note, that the reallocation of the angular distribution of the light has a different origin, than the remaining inhomogeneity at the SH exit plane. The achieved near-field homogenization is dependent on the remaining interference fluctuations in the SH transfer function. In contrast, the variations in the spectrograph illumination are based on a geometrical reallocation of the angular distribution of the light exiting the SH in combination with interference effects in the spectrograph pupil plane.

In the following we make the geometrical argument rigorous using diffraction theory. A general case for the connection between slit exit plane and spectrograph pupil plane is considered by Goodman (2005, p. 104). In the scenario discussed there, a collimated input field $U_l(x_s, y_s)$ propagates through a perfect thin lens at a distance d. The field in the focal plane of the lens is then given by:

$$U_f(u_b, v_b) = \frac{1}{i\lambda f} \, exp\left( i\frac{k}{2f} \left( 1 - \frac{d}{f} \right) (u_b^2 + v_b^2) \right) \int\limits_{-\infty}^{\infty} \int\limits_{-\infty}^{\infty} U_l(x_s, y_s) \, exp\left( -i\frac{k}{f} (x_s u_b + y_s v_b) \right) dx_s dy_s \qquad (5)$$

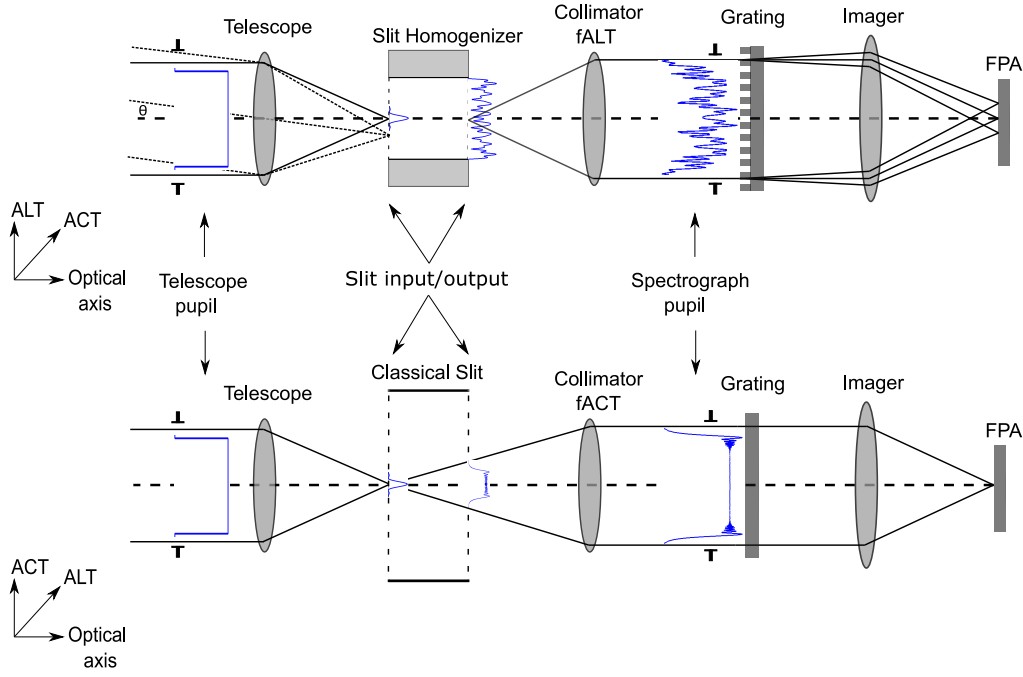

**Figure 4.** Generic setup of the SH in the Sentinel-5/UVNS instrument. A plane wavefront gets focussed in the SH entrance plane and the propagation of such stimulus is shown in blue as the square modulus of the electric field. The incoming light undergoes several reflections in ALT direction, whereas the SH in ACT is similar to a classical slit acting as a field stop. The collimator contains an astigmatic correction which is adjusted to the slit length. The SH homogenizes the scene in ALT direction but also modifies the spectrograph pupil illumination. The grating disperses the light in ALT. The pupil distribution in ACT direction is conserved except for diffraction effects due to truncation of the telescope PSF in the slit plane.

where $x_s, y_s$ are the position in the spectrometer pupil plane and $u_b, v_b$ the coordinates in the image plane at the SH exit. Indeed, the field at the lens focal plane is proportional to the two-dimensional Fourier transform. In contrast, our situation is inverted as we are interested in $U_l(x_s, y_s)$, i.e. the collimated field distribution at the spectrometer pupil originating from the SH output plane. Further, we need to incorporate the astigmatism in the collimation optics and the diffraction grating. These steps are covered in the following two sections.

## 2.3 Collimator astigmatism

In order to keep the full image information in ACT while imaging the homogenized SH output image, the collimator needs an astigmatism. In our model, this is implemented via Zernike polynomial terms on the collimation lens. We follow the OSA/ANSI convention for the definitions of the Zernike polynomials and the indexing of the Zernike modes (Thibos et al., 2000). The focal length of the collimator in ALT is such to image the SH exit plane, while in ACT the SH entrance plane is imaged. In the

simulation this is realised with three terms: a focal length term where the focal length is that of the collimator in ALT, a defocus term to shift the object plane and an astigmatism term to separate the ALT (tangential) and ACT (saggital) object planes.

The Zernike polynomials are given by:

Defocus: $\quad Z_2^0(\rho,\theta) = c_{02}\sqrt{3}(2\rho^2 - 1)$ (6)

Astigmatism: $\quad Z_2^2(\rho,\theta) = c_{22}\sqrt{6}\rho^2 sin(2\theta)$ (7)

where $c_{nm}$ are the Zernike coefficients, defining the strength of the aberration and $Z_n^m$ the Zernike polynomials. Due to the elegant and orthonormal definition of the Zernike polynomials, a perfect matching of Defocus and Astigmatism amplitude is straightforward, as the difference between the sagitta and tangential plane of the astigmatism is solely dependent on the radial term of the Zernike polynomial. Therefore, in order to match the corresponding difference given by the SH length, the weighting of the astigmatism has to be larger than the defocus term by a factor of $\sqrt{2}$. Hence, the combined Zernike term will be:

$$H(\rho,\theta) = c\, Z_2^0(\rho,\theta) + \sqrt{2}\, c\, Z_2^2(\rho,\theta)$$ (8)

Including the astigmatism of the collimation optics, applying $d = f_{col,ALT}$ and solving eq. (5) for $U_{l,\theta}$ by using the coordinate transformation $x_s' = \frac{k}{f}x_s$ and $y_s' = \frac{k}{f}y_s$, we get the field distribution at the diffraction grating as:

$$U_{l,\theta}\left(x_s', y_s'\right) = \frac{i}{\lambda f}e^{ik\left(c\, Z_2^0(\rho,\theta) + \sqrt{2}\, c\, Z_2^2(\rho,\theta)\right)}$$

$$\cdot \int_{-\infty}^{\infty}\int_{-\infty}^{\infty} U_{f,\theta}(u_b,v_b)\ exp\left(i\frac{k}{f}(x_s'u_b + y_s'v_b)\right)du_b dv_b$$ (9)

Equation 9 yields the field distribution incident on the diffraction grating. The implementation of the diffraction grating, which is responsible for the wavelength dispersion will be introduced in the next section.

## 2.4 Diffraction grating

The primary goal of the spectrometer is to distinguish the intensity of the light as a function of the wavelength and spatial position. In order to separate the wavelengths a diffractive element is placed in the spectrograph pupil and disperses the light in the ALT direction. For our analysis, we place the diffraction grating at a distance $d = f_{col,ALT}$ after the collimator and on the optical axes. Further, we model the dispersive element as a 1D binary phase diffraction grating. Such gratings induce a $\pi$ phase variation by thickness changes of the grating medium. Three design parameters are used to describe the grating and are unique for every spectrometer channel: the period of the grating $\Lambda$, the phase difference $\Phi$ between the ridge (of width $d$) and the groove regions of the grating, and the fill factor $d/\Lambda$. Physically, the phase difference itself is induced by two parameters: the height or thickness $t$ of the ridge and the refractive index of the material of which the grating is made. In most cases, the

refractive index of the used material is fixed and the thickness of the material is the primary parameter. The phase profile with a fill factor of 0.5 which provides the maximum efficiency in ALT direction is given by:

$$\Phi_{1D}(y_s) = \begin{cases} \pi & 0 \leq y_s \bmod \Lambda \leq \frac{\Lambda}{2} \\ 0 & \frac{\Lambda}{2} \leq y_s \bmod \Lambda \leq \Lambda \end{cases} \tag{10}$$

The complex electric field of the spectrograph pupil wavefront after the diffraction grating is then given by:

$$U_{g,\theta}(x'_s, y'_s) = U_{l,\theta}(x'_s, y'_s) \, e^{i\Phi(y_s)} \tag{11}$$

The intensity distribution after the grating is given by inserting equation (9) in (11) and applying the absolute square:

$$I_{g,\theta}(x'_s, y'_s) = |U_{g,\theta}(x'_s, y'_s)|^2 \tag{12}$$

The implementation of the diffraction grating is a simplified model, which is an approximation of the real, more complex
case. In Sentinel-5/UVNS, the SWIR spectrograph is equipped with a silicon immersed grating. The simplified approach is also valid for this case, as the SH does not affect the general behaviour of the grating.

## 3  Spectrograph pupil intensity distribution

The far-field intensity distribution is dependent on the contrast of the Earth scene in ALT and therefore on the SH entrance plane illumination. We characterize the amplitude of the variations of the spectrograph pupil illumination by introducing two
types of heterogeneous scenes. First, an applicable Earth scene as defined by ESA for the Sentinel-5/UVNS mission, which aims at representing a realistic Earth scene case. The on ground albedo variations of this scenes can be parametrized as a linear interpolation between two spectra, representing the same atmospheric state, but obtained with either a dark or bright albedo (Caron et al., 2017). The spatial variation of the scene heterogeneity is described by introducing interpolation weights $w_k$. The resulting spectrum for a given ALT subsample $k$ is then calculated as:

$$L_k(\lambda) = (1 - w_k) \, L_{dark}(\lambda) + w_k \, L_{bright}(\lambda) \tag{13}$$

where the reference spectra correspond to a Tropical bright scene ($L_{bright}$ - albedo = 0.65) and a Tropical dark scene ($L_{dark}$ - albedo = 0.05). The weighting factors that were used for this study have been derived from the Moderate Resolution Imaging Spectroradiometer (MODIS) surface reflectance products with 500 m spatial resolution and total coverage of 25 km for relevant conditions of Sentinel-5/UVNS (EOP PIO, 2011). The slit smearing due to platform movement is accounted for by convolving
the on ground scene with the motion boxcar of the spatial sampling distance (SSD). The platform movement is acting like a low-pass filter and averages out short albedo variations with respect to the SSD and the instruments FoV. However, without a SH, remaining inhomogeneities are present in the slit which yield up to 20 % slit illumination variations in ALT directions. Figure 5 depicts the on ground albedo contrast given in terms of weighting factors $w_k$, the scene after smearing due to the motion of the platform and the location of the SH entrance plane. We assume the scene to be homogeneous in ACT direction.

In fact, heterogeneous scenes in ACT direction may also affect the ISRF stability in the presence of spectrometer smile (see Gerilowski et al. (2011) and Caron et al. (2017)).

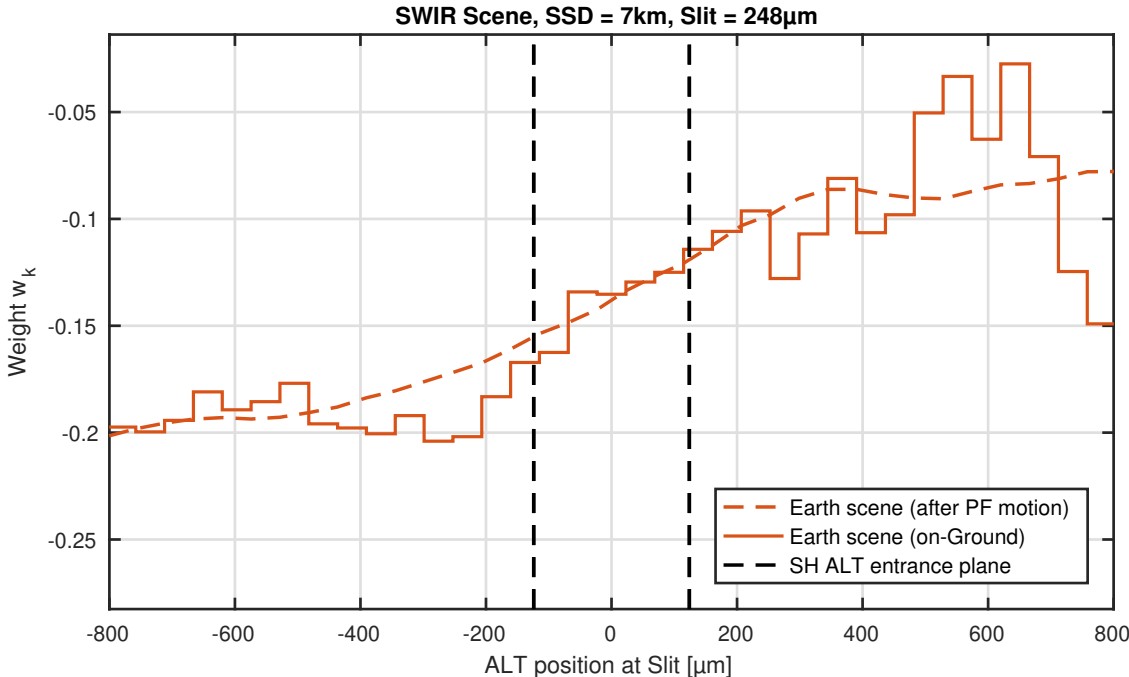

**Figure 5.** Realistic Earth scenes in the SWIR-3 derived from MODIS images corresponding to the slit illumination in ALT. The on ground surface albedo is given in terms of weight factors $w_k$ in the solid line. The same scene after smearing with a boxcar of the spatial sampling distance (SSD) accounting for the platform motion is given in the dashed line. The scene contrast including the platform motion in the plane of the SH entrance plane will be the reference scene for this study.

The second scene considered represents an artificial calibration (CAL) scene where $50\%$ of the slit is illuminated and $50\%$ is dark. These kind of instantaneous transitions are impossible to be observed by a push-broom instrument with finite FoV and integration time. However, they are convenient to be applied in experimental measurements and will serve as reference to
235 experimentally validate the SH performance models.

Figure 6 depicts the simulation results for the pupil intensity distribution in the SWIR-3 (2312 nm) for the applied test scenes as well as a homogeneous slit illumination. As expected, the uniformity of the input telescope pupil illumination is completely conserved in ACT direction due to the absence of interaction, i.e. reflection, with the SH. Therefore the top-hat intensity distribution of the telescope is, besides diffraction edge effects, completely preserved. To the contrary, the intensity distribution
in ALT is dependent on the contrast of the applied scene. Even for a homogeneous scene the SH modifies the pupil intensity (Fig. 6a) and consists of symmetrical variations. The intensity pattern just varies slightly for the applicable Earth scene (Fig. 6b ) due to the moderate gradient of the slit illumination variation. The CAL scenes (Fig. 6 c,d) highlight the previously made

geometrical argument for the non-uniform pupil illumination as parts of the pupil are left with only a fraction of the light. For illustration, we show a case where the upper 50 % of the slit are illuminated and another case where the lower 50 % of the slit are illuminated (representing the ALT illumination).

In the next section we will investigate the impact of non-uniform pupil illumination in combination with spectrograph aberrations on the ISRF stability.

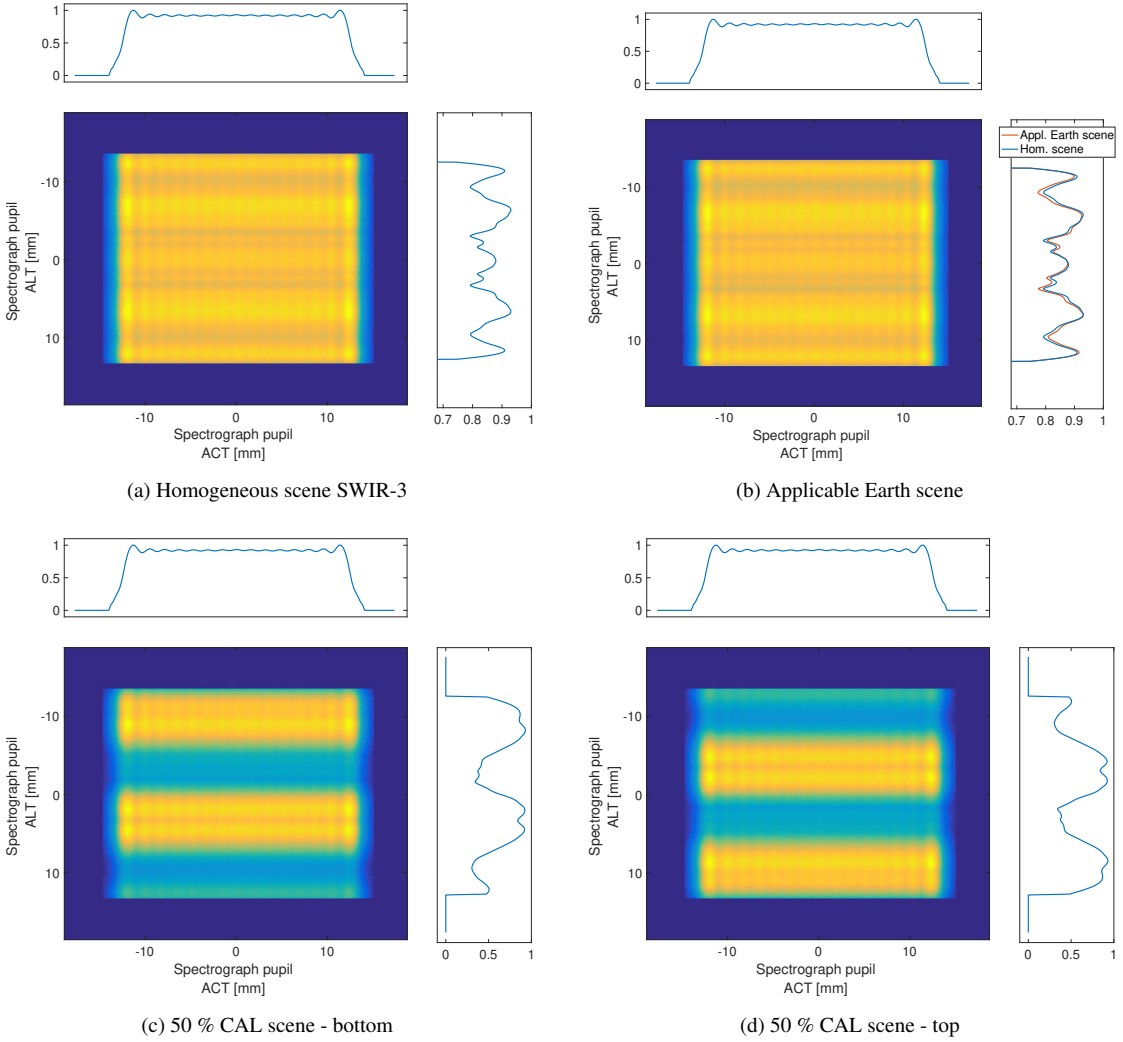

**Figure 6.** Simulation results of the spectrograph pupil intensity distribution in the SWIR-3 (2312 nm) for different slit illuminations. The uniformity of the pupil in ALT is dependent on the applied scene. The ACT uniformity from the telescope pupil is preserved, as there is no interaction with the SH.

## 4 Impact on ISRF

The main impact of the above described variations in the spectrometer pupil illumination is the scene dependent weighting of the aberrations inherent to the spectrograph optics. In the case of a classical slit, it is valid to calculate the ISRF of an imaging spectrometer as the convolution of the slit illumination, the pixel response on the FPA and the optical PSF of the spectrograph optics. When using a SH, a scene dependency of the spectrograph pupil illumination will weight the aberrations of the system accordingly and thereby create a variation in the PSF, which will ultimately also change the ISRF properties. Therefore, it is necessary to keep the complex phase of the electric field during the propagation through the instrument.

Instead of a convolution, we propagate the spectrograph pupil illumination through the imaging optics by diffraction integrals. For the description of the aberrations present in the Sentinel-5/UVNS instrument we use again the formulation of Zernike theory. We know the expected PSF size on the FPA of the Sentinel-5/UVNS SWIR-3 channel, which in the case of a classical slit can be approximated by the standard deviation of a normal distribution. In order to assess the impact of aberrations, we impinge different types of aberrations on the spectrograph imaging optics and match the PSF size to the instrument prediction. As the shape of the PSF for an arbitrary aberration is not given by a normal distribution, we define the PSF size as the area where $80\,\%$ of the encircled energy (EE) is contained. Then we tune the strength of the aberration coefficients in such a way that the size of the aberrated PSF matches that of the normal distributed PSF. For the transformation of the spectrograph pupil illumination to the FPA including aberrations, we apply the thin lens formula and expand it by adding the phase term for the Zernike aberrations (Goodman, 2005, p. 145). Our starting point for the propagation is the grating position where, for the case of Sentinel-5/UVNS, the distance $d$ is matching the focal length of the imaging optics. In that case the formulation simplifies again and is given by a relation which has the form of a Fourier transform:

$$U_{FPA,\theta}(s,t) = \frac{1}{i\lambda f_{im}} \int\limits_{-\infty}^{\infty} \int\limits_{-\infty}^{\infty} U_{g,\theta}(x'_s,y'_s) \, exp\left(-i\frac{k}{f_{im}}(x_s s + y_s t)\right) exp\left(\frac{ik}{\pi} H(r,\phi)\right) dx_s dy_s \tag{14}$$

where $s,t$ are the coordinates at the FPA, $f_{im}$ is the focal length of the imager, $U_{g,\theta}$ the field distribution at the grating and $H(r,\phi)$, with $r = r(x_s,y_s)$ and $\phi = \phi(x_s,y_s)$, the respective Zernike aberration that we apply. Any spatially incoherent monochromatic input scene can be distributed in plane wavefronts with amplitude $A(\Theta)$. Each such wavefront leads to an intensity $I = I_\Theta(s,t) = |U_{FPA,\theta}|^2$ on the FPA. As we have no SH impact in ACT direction, we collapse this dimension and sum along it. This yields the 1D ISRF intensity distribution on the FPA as a function of the incidence angle $\Theta$ as $I_\Theta(t)$. The respective scene will weight the intensities on the FPA depending of their strength and is therefore the linear operator:

$$I_t = \int\limits_{\Theta \in \mathbb{R}} A(\Theta) I(\Theta,t) \, d\Theta = I \circ A(t) \tag{15}$$

Note that for a homogeneous scene, $A(\Theta) = 1$ for every incidence angle. Finally, the normalized ISRF on the FPA is given by:

$$\widetilde{ISRF}(t) = (I_\Theta \circ A) * \chi * N_\sigma(t) \tag{16}$$

$$ISRF(\lambda) = \frac{\widetilde{ISRF}(\frac{\lambda}{\alpha})}{\alpha \int \widetilde{ISRF}(t) \, dt} \tag{17}$$

where $\chi$ is the characteristic function, which is 1 inside a pixel area and 0 elsewhere, $\alpha$ a scaling factor to give the ISRF in units of wavelength ($\lambda$) and $N_\sigma$ is the density function of a normal distribution with zero mean value and standard deviation $\sigma$. The latter factor accounts for the modulation transfer function (MTF) of the detector (not the MTF of the whole optical system).

In order to asses the stability of the ISRF we define three merit functions:

- Shape error, which we define as the maximum difference of the ISRF calculated for a homogeneous and heterogeneous scene respectively

$$\text{Shape error} := \max_{\lambda} \left| \frac{ISRF_{hom}(\lambda) - ISRF_{het}(\lambda)}{\max_{\tilde{\lambda}} ISRF_{hom}(\tilde{\lambda})} \right| \tag{18}$$

- Centroid error: Shift of the position of the spectral channel centroid, where the centroid is defined as

$$\text{Centroid error} := \frac{\int\limits_{FPA} ISRF(\lambda)\, \lambda \, d\lambda}{\int\limits_{FPA} ISRF(\lambda)\, d\lambda} \tag{19}$$

- Spectral resolution of the ISRF given by the FWHM

We consider two cases for the assessment of the induced impact on the ISRF stability. In the first case, we neglect any variation of the spectrograph illumination and use the PSF as a convolution kernel of the ISRF given as a constant and scene independent normal distribution defined as:

$$g(t) = \frac{1}{\sigma\sqrt{2\pi}}\, exp\left( -\frac{t^2}{2\sigma^2} \right) \tag{20}$$

where $\sigma$ is the standard deviation representing the size of the PSF. The spot size value for a representative field point in the SWIR-3 spectrometer of Sentinel-5/UVNS is about $6.85$ $\mu$m. When convolving with a gaussian PSF, we neglect the non uniformity in the pupil and the spectrometer aberrations and the ISRF errors are only driven by the slit exit illumination (near-field). For the second case, we impinge a certain amount of aberrations on the imaging optics to get the same spot size for the PSF as in the first case. In this case, the ISRF errors are a combination of the remaining inhomogeneities at the SH exit plane (near-field), as well as effects due to non-uniform spectrograph illumination (far-field). The aberrations present in the Sentinel-5/UVNS spectrograph are dependent on the position on the FPA in spectral and spatial direction. In the upcoming characterization and calibration campaign, the specific types of aberration of the final instrument will not be determined, but only the size of the spots. Therefore, although it is not a realistic case, we impinge pure aberrations of a single type in order to determine critical Zernike terms for the ISRF stability. We also test two mixtures of different types of aberrations, which represent more realistic field points of Sentinel-5/UVNS. The ISRF for a homogeneous scene including aberrations, will be extensively characterized on-ground. We want to investigate how the ISRF based on several Zernike terms behave under the condition of non-uniform scenes and how the ISRF deviation evolves with respect to each, aberration type specific, homogeneous ISRF. Therefore, in the next paragraph, we calculate the relative change in the ISRF figures of merit functions.

## 5 Results and discussion

In the following, we present the ISRF figures of merit resulting from the simulation of several Zernike polynomials for the Sentinel-5/UVNS applicable heterogeneous Earth scene and a 50 % stationary calibration scene. Further, we compare the results to the case of a classical slit without scene homogenization. Table 1 & 2 summarize the results for the ISRF figures of merit. Note that the errors for the calibration scene are much larger than the errors for a realistic Earth scene. The calibration scene can be used in a laboratory to characterize the SH performance and compare it with the prediction. All Zernike polynomials increase the error in the ISRF knowledge compared to the case, where the ISRF is calculated as the convolution with a constant gaussian PSF. The error magnitude variation ranges from only small increasing errors (Defocus, Vertical astigmatism) to a notable increase of the error (Oblique quadrafoil, Horizontal Coma). The aberrations change both the maximum amplitude of the errors and the specific shape of the ISRF. Figure 7b depicts the ISRF assuming pure vertical coma, pure spherical aberrations and pure oblique trefoil for a heterogeneous 50 % calibration scene. The lower part of the plot shows the ISRF shape difference for each specific homogeneous reference scene. Note, that the shape error is defined as the maximum amplitude of the difference plot. As none of the field points in the real Sentinel-5/UVNS instrument will contain a pure singular type of aberration, we tested two set of aberration mixtures, which is more representative of a real field point in the Sentinel-5/UVNS instrument. Although our study doesn't provide a rigorous mathematical argument, the results indicate, that the error of the combined Zernike polynomials lies within the errors of the individual contributors. This argument is supported by Fig. 8, where

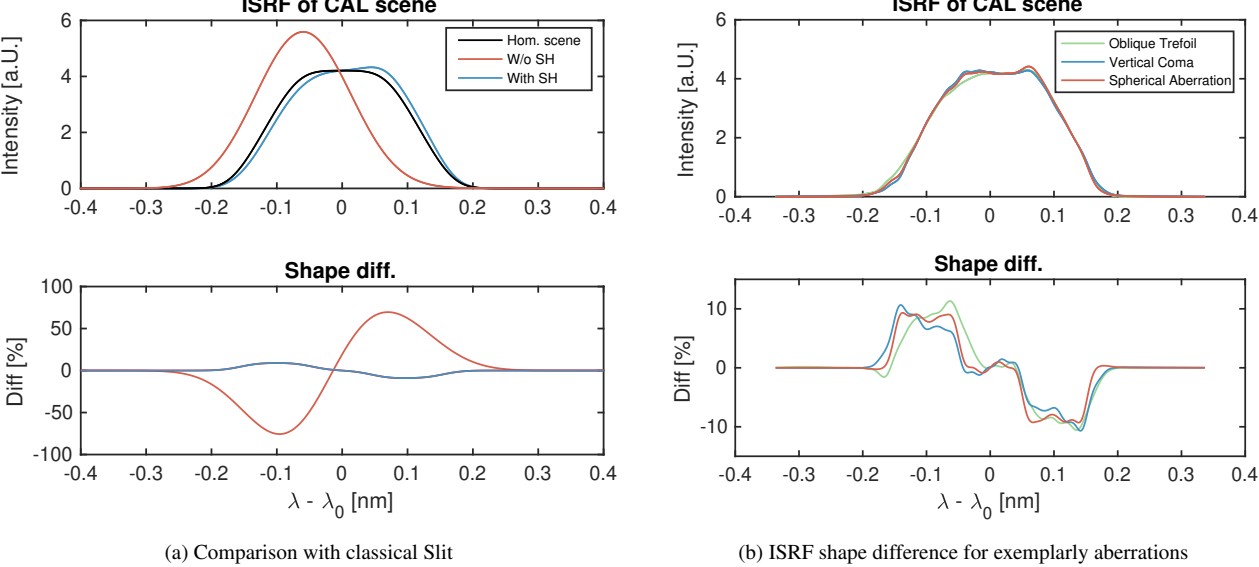

(a) Comparison with classical Slit  (b) ISRF shape difference for exemplarly aberrations

**Figure 7.** (a) ISRF with and without a slit homogenizer for a heterogeneous 50 % calibration scene. The SH strongly reduces the shape error of the ISRF by an order of magnitude. (b) Comparison between the ISRF shape errors for three exemplary aberrations. The presented aberrations induce a higher maximum shape error but also strongly change the overall shape of the ISRF with respect to the homogeneous reference case.

we plotted the ISRF shape error, going from a pure oblique quadrafoil aberration to a pure defocus aberration. In each step we reduced the fraction of the quadrafoil aberration by 20 % and tuned the defocus aberration coefficient in such a way, that we ended up with the same PSF size of 6.85 $\mu$m (80 % EE ). The ISRF errors always remain in the corridor between the case of pure oblique quadrafoil and pure defocus aberration. This behaviour was tested for several other Zernike combinations. From that we conclude, that the errors given in Table 1 & 2 for the respective Zernike polynomials span the error space, where mixtures of aberrations lie within.

Although the phenomena of the variations of the pupil illumination in combination with spectrometer aberrations increases

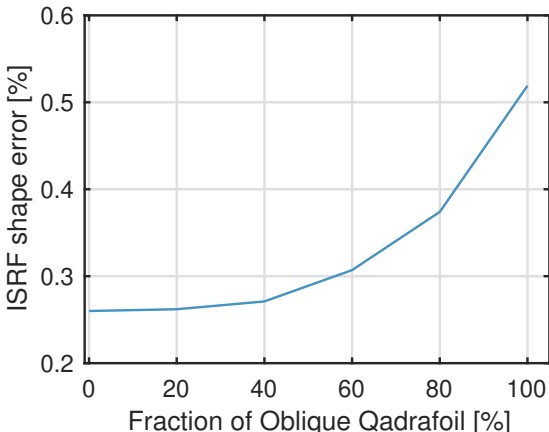

**Figure 8.** Progression of ISRF shape error from pure oblique quadrafoil aberration to pure defocus aberration. Between the values, we decreased the quadrafoil Zernike coefficient in 20 % steps and at the same time adjusted the defocus coefficient to reach the PSF design size of 6.85 $\mu$m again. The plot suggests that the ISRF errors of Zernike combinations are within the ISRF errors of the individual Zernike contributors.

the errors, the SH still homogenizes the scene well, and significantly improves the stability of the ISRF compared to a classical slit. In Fig. 7a we compare the ISRF shape difference for a 50 % stationary calibration scene for a case with a classical slit and a case with SH. The SH improves the ISRF stability by almost an order of magnitude. Considering the applicable Earth scene and including the far-field variations, the SH still provides sufficiently stable ISRF stability with respect to the mission requirements for moderate heterogeneous scenes of Sentinel-5/UVNS. This would not be the case for an instrument employed with a classical slit.

In certain scenarios, Sentinel-5/UVNS will fly over Earth scenes with higher contrasts than specified in the applicable Earth scene. This will be the case when flying over cloud fields, water bodies or city to vegetation transitions. However, these scenes are excluded from the mission requirements in terms of scene homogenization. Although sufficient for the purposes of Sentinel-5/UVNS, the capability of the SH to homogenize the scene is not perfect. This imperfection is particularly prominent when considering the calibration scenes. The imperfections originate from the remaining interference fluctuations in the SH transfer function and are dependent on the wavelength. Higher wavelengths show smaller frequencies and larger peak-to-valley ampli-

tudes of the maxima in the SH transfer function, which leads to reduced homogenization efficiency. Therefore, the SWIR-3 wavelength channel is the most challenging in terms of scene homogenization.

We observe, that increasing the number of reflections inside the SH will increase the number of stripes in the spectrometer pupil illumination (see Fig. 6c/6d) and reduce the peak to valley amplitude. This would lead to a more homogeneous pupil illumination. More reflection in the SH can be achieved by either increasing the length of the SH or adapting the telescope $F_\#$. However, it is advantageous to keep the SH length small to reduce the collimator astigmatism requirements. Note, that a longer SH would not increase the near-field homogenization performance. In addition, more reflections in the SH lead to greater transmission losses at the mirrors. As the errors due to the pupil illumination are small compared to achieved near-field homogenization, it seems favourable to prioritize the first-order design rule given in Caron et al. (2019) and Meister et al. (2017). The SH shows the best near-field homogenization performance if $F\#_{tel} = l/(2bn)$, where $F\#_{tel}$ is the telescope F-number, $l$ the SH length, $b$ the SH width and n the number of reflections. For Sentinel-5/UVNS, the optimal parameters for SWIR-3 are a telescope $F_\#$ of 9.95, a slit length of 9.91 mm and a slit width of 248 $\mu$m.

The simulation results of this study still require experimental validation. An initial approach to validate the SH transfer functions was published in Irizar et al. (2019), where they showed good agreement between the simulation and the experimental result for a single SH incidence angle. The verification of the full transfer function including the full FoV range is pending. The SH far-field effects investigated in this study could be determined by measuring the pupil intensity distribution at the grating position by means of an appropriate test bench. The test bench would need to be capable to illuminate the SH entrance plane through a telescope with angles representing the Sentinel-5/UVNS FoV. Further, the astigmatism of the SH needs to be compensated which could be done by introducing a cylindrical lens in the collimator system.

Apart from the mirror based SH discussed in this study, future remote sensing instruments investigate the technology of another slit homogenizer technology which is based on rectangular multimode fibre bundles. These devices are based on the same principle as the mirror based SH but enable to homogenize the scene in ACT and ALT direction (Amann et al., 2019) and provide enhanced performance over extreme albedo variations.

**Table 1.** Applicable Earth scene - ISRF stability. Requirements: Shape error $< 2$ %, FWHM error $< 1$ %, Centroid error 0.0125 nm. The presented errors combine the remaining SH exit non-uniformity (near-field) and effects due to the variations of the spectrograph pupil illumination (far-field). The strength of the aberrations are chosen such that the spot size matches the case of a PSF size of 6.85 $\mu$m (80 % EE).

| OSA/ANSI Index | Zernike Term | Shape Error [%] | FWHM Error [%] | Centroid Error [nm] |
|---|---|---|---|---|
| 3 | Oblique astigmatism | 0.344 | 0.056 | 0.0003 |
| 4 | Defocus | 0.260 | 0.023 | 0.0002 |
| 5 | Vertical astigmatism | 0.260 | 0.023 | 0.0002 |
| 6 | Vertical trefoil | 0.409 | 0.020 | 0.0002 |
| 7 | Vertical coma | 0.388 | 0.032 | 0.0003 |
| 8 | Horizontal coma | 0.490 | 0.055 | 0.0003 |
| 9 | Oblique trefoil | 0.451 | 0.103 | 0.0003 |
| 10 | Oblique quadrafoil | 0.519 | 0.017 | 0.0003 |
| 11 | Oblique second. astigmatism | 0.398 | 0.011 | 0.0003 |
| 12 | Primary spherical | 0.372 | 0.040 | 0.0003 |
| 13 | Vertical second. astigmatism | 0.382 | 0.040 | 0.0003 |
| 14 | Vertical quadrufoil | 0.380 | 0.030 | 0.0003 |
| Mixture 1 - Defocus (33 %) / V. astig. (33 %) / Prim. sph. (33 %) | | 0.334 | 0.017 | 0.0002 |
| Mixture 2 - O. astig (36 %) / V. coma (32 %) / O.s. astig (32 %) | | 0.382 | 0.040 | 0.0002 |
| With SH - Gaussian PSF | | 0.248 | 0.010 | 0.0003 |
| Classical Slit - Gaussian PSF | | 2.54 | 0.061 | 0.0030 |

**Table 2.** 50 % CAL scene - ISRF stability. The presented errors combine the remaining SH exit non-uniformity (near-field) and effects due to the variations of the spectrograph pupil illumination (far-field). The strength of the aberrations are chosen such that the spot size matches the case of a PSF size of 6.85 $\mu$m (80 % EE). Remark: ISRF values are exaggerated with respect to real flight scenarios. Calibration scenes are used for on-ground SH performance validation.

| OSA/ANSI Index | Zernike Term | Shape Error [%] | FWHM Error [%] | Centroid Error [nm] |
|---|---|---|---|---|
| 3 | Oblique astigmatism | 8.507 | 1.589 | 0.008 |
| 4 | Defocus | 6.883 | 0.884 | 0.004 |
| 5 | Vertical astigmatism | 6.883 | 0.884 | 0.004 |
| 6 | Vertical trefoil | 9.230 | 0.833 | 0.008 |
| 7 | Vertical coma | 9.320 | 2.025 | 0.008 |
| 8 | Horizontal coma | 11.549 | 2.250 | 0.008 |
| 9 | Oblique trefoil | 11.320 | 0.566 | 0.008 |
| 10 | Oblique quadrafoil | 11.859 | 3.316 | 0.008 |
| 11 | Oblique second. astigmatism | 10.059 | 3.750 | 0.008 |
| 12 | Primary spherical | 10.686 | 0.382 | 0.008 |
| 13 | Vertical second. astigmatism. | 11.136 | 0.465 | 0.008 |
| 14 | Vertical quadrufoil | 10.127 | 0.928 | 0.008 |
| Mixture 1 - Defocus (33 %) / V. astig. (33 %) / Prim. sph. (33 %) | | 7.367 | 0.442 | 0.004 |
| Mixture 2 - O. astig (36 %) / V. coma (32 %) / O.s. astig (32 %) | | 9.982 | 0.849 | 0.008 |
| With SH - Gaussian PSF | | 6.363 | 0.566 | 0.008 |
| Classical Slit - Gaussian PSF | | 65.664 | 37.039 | 0.059 |

## 6 Conclusion

The presented study continues the investigation by Caron et al. (2019) and Meister et al. (2017) on the mirror based slit homogenizer technology. While the preceding studies were considering the homogenization of the SH exit plane, here, we extend the models by including the electric field propagation through the subsequent spectrograph. The slit homogenizer not only homogenizes the slit illumination, but also modifies the spectrograph illumination dependent on the input scene. The variations in the spectrograph pupil illumination will lead to a scene dependent weighting of the geometrical aberrations in the optical system, which cause an additional distortion source of the ISRF. The phenomena is particularly prominent in the presence of extreme on-ground albedo contrasts. This will be the case, when the instrument flies over clouds or water bodies. However, in the context of the Sentinel-5/UVNS instrument, these scenes are excluded from the mission requirements.

We observe, that the impact of spectrograph pupil illumination variations is small compared to the error due to non-uniform slit illumination and the ISRF distortion is primary driven by the remaining near-field variations after the SH. The inhomogeneity remnants arise from the fluctuations of the interference pattern at the SH exit plane. The strength of the variations is increasing with wavelength. Therefore, this study was conducted in the SWIR-3 channel in order to cover the worst case.

We quantify the ISRF in terms of shape error, FWHM error and centroid error at 2312 nm by an end to end propagation through the SH and the subsequent spectrograph optics. With regard to these figures of merits, our simulation results suggest an increase of the errors depending on the specific type of aberrations impinged on the optics. ISRF errors of combined Zernike polynomials are always within the maximum errors of the individual Zernike constituent. Although the SH changes the spectrometer illumination, it still has significant performance advantages in stabilizing the ISRF compared to a classical slit. For an applicable heterogeneous Earth scene, the SH improves the ISRF shape stability by a factor of 5-10. The remaining residual errors are well below the Sentinel-5/UVNS system requirement, which are: shape error $2\%$, the relative FWHM error $< 1\%$ and the centroid error $< 0.0125$ nm (SWIR-3).

*Data availability.* The datasets generated and/or analyzed for this work are available from the corresponding author on reasonable request, subject to confirmation of Airbus Defence and Space GmbH.

*Author contributions.* Timon Hummel was responsible for the modelling and the analysis of the simulation data supported by all co-authors. Timon Hummel developed the end-to-end model and the approach to quantify the impact on the ISRF supported and revised by Christian Meister. Timon Hummel prepared the manuscript with contributions and critical revision from all co-authors.

*Competing interests.* The authors declare, that they have no conflict of interest.

*Acknowledgements.* We thank Tobias Lamour, Jess Köhler and Markus Melf (all Airbus Defence and Space) for the helpful comments on a previous version of the manuscript.

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
