# Peer review of "Slit homogenizer introduced performance gain analysis based on Sentinel-5/UVNS spectrometer"

_Atmospheric Measurement Techniques, 2021_

## Author Response (AR1)

**Author response to the revised manuscript version**

**June 18, 2021**

This document repeats the questions of all referees with the corresponding authors' responses on a point to point basis. Additionally, it indicates for every comment the changes made in the manuscript and explains the reasons of the authors if needed. All page and line references refer to the revised manuscript version. We introduced a numbering of the comments and hope that it reflects the referees view. Appended to this document is a differential view between the first and revised version for convenient tracking of the applied changes.

**1 Comments Referee 1**

**1.1 General Comments**

R2: The authors state at the end of section 4 that the "discrepancy in the [ISRF] values is quite significant" and that "we believe that depending on the mission parameters, this effect should be taken into account for the assessment of the ISRF stability and consequently the performance of the SH". But then in the next sentence they state: "We also conclude, that for the Sentinel-5/UVNS instrument the impact of this effect is of second-order and does not degrade the performance of the SH significantly". This important conclusion is however stated without any further motivation or evidence. It also seems contradictory to the previous sentence. In contrast, the error budget from tables 1 to 3 should be discussed in view of the S5 ISRF requirements error budget, which is intimately linked to the Sentinel-5 product requirements and quality. In this respect, the nature of the S5-ESA scene should be discussed. Is this scene referring to the type 2 non-uniform scene as defined by the S5 system requirements document (Appendix A)? While this is meant to represent a realistic scene with inhomogeneities representing a more averaged land situation, the still moderate and more randomly distributed signal variations result in quite uniform smeared out signal conditions in along-track direction (averaged over the 7km across-track footprint of S5). So the 75% scene presented her seems to be a more realistic case for typical non-uniform scenes, with sharp surface type transitions (city or desert to vegetation, or land to water). The latter seems never to meet the 2%ISRF shape error budget of the S5 SRD not even for a normally distributed PSF.).

Response: We confirm that the S5-ESA scene is referring to the type 2 non-uniform scene as defined by the S5 SRD. We will describe the derivation of this scene in more detail.

In our manuscript, the calibration scenes refer to conditions with a sudden transition from bright to dark illumination without accounting for motion smear of the satellite platform. These scenes will be used for static on-ground laboratory measurements of the slit homogenizer performance and to validate the prediction models. We agree, that we should make the use case of these scenes more clear and that the ISRF distortions associated with the calibration scenes are not representing real flight measurements but will only be measured in the laboratory. The resulting ISRF errors are exaggerated with respect to real in-flight scenarios. In the revised version of the manuscript we will describe the realistic scenes from the SRD and make our performance assessment based on these realistic scenes. We will only keep an exemplary 50% stationary calibration scene result and emphasize their use case and the resulting exaggerated ISRF distortions.

The aberrations present in the Sentinel-5/UVNS spectrograph are dependent on the position on the FPA in spectral and spatial direction. Further, the specific aberration type of the final instrument will not be determined but only the RMS spot sizes. In a revised manuscript we will consider several other types of aberration and also their mixing behaviour to make a more thorough and realistic case of the Sentinel-5/UVNS spectrograph.

Changes in manuscript: In section (3), we added a derivation and explanation of the origin of the applicable Earth scene for the Sentinel-5/UVNS mission as defined by ESA in the system requirement document (SRD). We put the calibration scenes into perspective, as they are only used in laboratory measurements. Further, we will mention that measurements over extreme albedo variations are excluded from the Sentinel-5/UVNS mission requirements as stated in the SRD.

**R2: 1. I think it would be interesting to also add the expected ISRF error for an optics without SH to the results (tables) presented in Section 4, if that would be possible. Since this would provide the reference with respect to the currently flying push-broom missions.**

Response: We will include simulation results of the case with no slit homogenizer present in order to compare with push-broom missions using a classical slit.

Changes in manuscript: The simulation results for a case of a classical slit are presented in Table (1) and (2). In Figure (7a), we show an ISRF comparison plot with and without SH for a 50 % calibration scene.

R2: 2. The reasoning for making the case for slit-homogenizations, as presented in the context of future missions with even higher spatial resolution like CO2M (Section 3, line 195ff), is a bit confusing. Although I understand, what the authors intend here. The relevance for CO2M is not in terms of CO2 emission inhomogeneities, but again, as for the other missions, in terms of radiances variation. The latter is in the extreme cases governed by clouds and surface and not dominated by atmospheric constituents. Especially the variation of CO2 emission is at times at the sub percent level to the background, therefore not contributing to radiance scene homogeneities. However, underlying variations in surface reflection (e.g. transitions of cities to rural land and lakes) may cause significant ISRF distortions without proper slit-homogenizations, which then, in turn would affect the very high accuracies needed to quantify the elevated CO2 emission plume concentrations. So in this respect NO2 emissions may provide a better example of a single point variations, although even there I would assume that the largest effect on NO2 retrieval accuracies due to ISRF distortions is still originating from surface variations or cloud edges.

Response: We agree and will revise this section.

Changes in manuscript: The references to the CO2M mission have been removed from the manuscript and we discuss the analysis solely with respect to Sentinel-5/UVNS.

**1.2** Editorial comments**

R2: 1. Section 1, line 34ff: I would add here the linear detector array spectrometer with scanning mirrors like GOME-1/2 and SCIA have a large IFOV in along-track direction and a box-cart like PSF. You could also mention GOME-2 [Munro et al.,

2016] in this respect.

Response: We will add this information.

Changes in manuscript: see lines 44-46.

**R2: 2. Section 2.1, line 106. Shouldn't this reference be to Fig. 3b and not a?**

Response: Indeed, this will be changed.

Changes in manuscript: see line 122.

R2: 3. Section 2.2, line 128: missing space.

Response: Done.

Changes in manuscript: see line 149.

**2 General comments Referee 2**

**2.1 General comments/Specific Comments**

R2: 0. It is understood that pupil inhomogeneity is introduced by far-field interference originating from wave propagation within the mirror-based SH (creating multiple coherent sources), and would not occur with a classical slit. The far-field nonuniformities are also independent from the achieved near-field homogenisation, and are not a consequence of "remaining" inhomogeneities. This should be stated more clearly in the text (both, if this understanding is correct and if not).

Response: In general, the near-field and far-field after the SH are correlated by Fourier transform as stated in this paper. But the reasons for the non-uniformity have different origins. The remaining inhomogeneities at the SH exit plane are a result of interference effects of the light propagating through the SH in combination with diffraction. However, the non-uniformity of the pupil illumination originates from the reallocation of the angular distribution of the light exiting the SH in combination with interference effects in the pupil plane. We will state this more clearly in the revised version.

Changes in manuscript: see lines 155-159.

R2: 1. Limitation of the analysis to a single wavelength in a single band, which is insufficient to draw far-reaching conclusions on the performance of wide-bandwidth multiplespectrometer instruments.[...]

While the methodology and the developed model seem adequate to analyze the impact of using a one-dimensional slit homogeniser, the present analysis is rather limited, and should be extended to come to more meaningful results w.r.t Sentinel-5 and potentially other space missions. The single analysed wavelength is in the NIR band of Sentinel-5, but Fig. 2 shows the SWIR-3 spectrum (with CO, CH4 and H2O absorption). The SWIR-3 is likely the most critical band in terms of the impact on retrieved products (e.g. CH4 and CO column densities). This is due to the deep absorption structures and the relatively high spectral resolution (although the O2 A-band in the NIR may also be critical). It is therefore recommended to extend the analysis at least to the SWIR-3 band at 2.3 µm (and ideally also SWIR-1 at 1.6µm). The mathematical model should still be valid for these spectral ranges, and the results would give an impression on the wavelength-dependency of pupil inhomogeneity and its impact on the ISRF. At minimum, please explain why the analysis was not performed for the SWIR-3 band plotted in Fig. 2. In this case it is suggested to replace the plot by the NIR. Another important extension of the analysis would the application to other wavelengths within the chosen band. The main impact of ISRF distortion (or knowledge error) is due to its variation within the spectral band used for the retrieval of the targeted molecular species. The results for one wavelength (or spectral channel) at 760nm reported here give no insight in intra-band variability. However, ISRF knowledge is required over the entire spectral range. Please discuss the expected variation of the results with wavelength. Do they only differ in terms of the contrast, which is lower in deep absorption lines?Do the errors in the figureof-merit (shape, FWHM, and centroid position) scale linearly with contrast?

Response: We agree, the analysis that we presented only considers a single wavelength point in the NIR channel which contains only limited information about the general SH performance and as pointed out by the referee is also not the most critical one in terms of the retrieved products. However, it is not our intention in this work to provide a full validation of the SH for the Sentinel-5/UVNS missions which will be part of the characterization and calibration campaign. Instead, we wanted to investigate phenomenologically the impact of a non-uniform spectrograph illumination and provide a simulation approach to assess the impact on the ISRF. We therefore propose to present the analysis in the SWIR-3 band instead of the NIR for two reasons. First, as already mentioned this is the more interesting channel in terms of retrieval products. Second, the SH is known to show better performance for smaller wavelengths, as the peak to valley variations in the SH transfer function are stronger for higher wavelengths. This will result in a reduced homogenization performance.

The far field effects that we investigate are driven by geometrical arguments and diffraction, which only vary slightly in a single wavelength channel. Therefore, the expected changes of intraband variability are small. This is particularly the case for the SWIR-3 channel in which the bandwidth is small compared to the wavelength.

Changes in manuscript: The entire manuscript has been adapted to the SWIR-3 wavelength channel.

 $R_2$ : 2. Limitation to four artificial input scenes, none of which representing a realistic in-flight scenario (even though one is referred to as "representative Earth scene").[...] The S5-ESA-scene used in this study, on which the compliance statement for Sentinel-5 is based on, remains obscure, in the sense that its origins and generation remain unclear. Although it is referred to as "representative Earth scene", the very low contrast plotted in figure 5 does not appear realistic for a ground-scene with considerable contrast. The instrument is likely to frequently see much higher contrast in orbit, e.g. when flying over cloud fields (bright in the NIR) or water bodies (dark in the SWIR). It is suspected that the authors picked one wavelength (in the continuum of the NIR band) of an artificial contrast scene from the S5 requirement documents, and convolved a brightdark step transition with the motion boxcar of the ALT spatial sample (please confirm or not). This is however not flight-representative, as such artificial reference scenes are typically designed to specify straylight performance, not to define a representative geophysical scenario. In order to demonstrate its relevance to expected Sentinel-5 performance, the authors shall clarify the origin and processing of the "S5-ESA-scene". What are the geophysical assumptions behind the scene ? Was it convolved to account for "motion smear" over the ALT spatial sampling distance? It is also recommended to extend the analysis to more than just one (basically flat) convolved transition, in stating compliance of the mission. The method would even allow to explore the maximum contrast transition, which would still lead to compliant ISRF knowledge requirements for the Sentinel-5 instrument. This would represent a relevant performance prediction for Sentinel-5, and greatly enhance the scope of the conclusions. The three other scenes considered (25%,50%, and 75% ALT slit illuminated) are also artificial (even impossible to be observed in flight). However they could serve the purpose of highlighting the criticality of pupil inhomogeneity for on-ground calibration. The authors should comment on the implications of their results for the on-ground ISRF characterization, which typically prescribes measurements with partially illuminated slit widths. At any rate, instantaneous transitions are impossible be observed by any pushbroom instrument with finite FoV and integration time. Therefore they cannot be claimed to be representative for so-called "high-contrast missions" (which is not a defined category anyway).

Response: We agree, the description of the scene is missing and the derivation of it is not mentioned yet. We will include a description of the applied realistic test scene case, how it is designed and how we accounted for the slit smearing due to the satellite motion. The data for the scene is referring to the Type-2 scene of the Sentinel-5/UVNS System requirement document (Sentinel-5 UVNS Instrument Phase A/B1 Reference Spectra) and represents the scenes, under which the SH has to reduce the contrast in alignment with the mission requirements. We agree that Sentinel-5 may observe scenes with higher contrasts, which are however excluded from the mission requirements. Therefore we propose to constrain our study to the mission requirements as defined by ESA and apply our SH model to the scenes defined in the SRD. The interpretation and justification of why the scenes defined in the SRD are representative of the Sentinel-5 mission are out of the scope of this paper.

We will revise the discussion on the stationary calibration scenes and how they are used in the Sentinel-5 project. As mentioned by the referee, the stationary calibration scenes are not a realistic scenario and will never be seen by the instrument in the presence of motion smear of the platform. We will mention the stationary calibration scenes for the purpose of on-ground SH performance verification.

Changes in manuscript: The derivation and explanation of the applicable Earth scene for the Sentinel-5/UVNS mission is given in section (3) (lines 214-232). Note that we changed the name from "S5-ESA-Scene" to "applicable Earth scene". The discussion of the application of the calibration scenes is also given in section (3) (lines 233-236). We also mention in the newly established "Results and discussion" section, that extreme albedo variations are excluded from the Sentinel-5/UVNS mission requirements (line 339) and in Section 6 (line 374).

**R2: 3. Unclear, somewhat arbitrary assumptions on imaging aberration, which partly invalidates the applicability of the results to the instrument under investigation (Sentinel-5).[...]**

For the propagation of the non-uniform pupil illumination pattern to the focal plane, aberration theory is used, describing the various types aberration with Zernike polynomials. While this is an adequate approach, the derivation of Zernike coefficients appears odd. It is claimed that the aberration components of the Sentinel-5 instrument are unknown, which seems surprising given the long history of design and development work (starting 2010). Instead, two aberration types are presented (spherical and comatic), without further explanation why they are selected to represent the instrument. The coefficients are derived by fitting the width of the PSF to the expected Gaussian (from optical analysis), implying that only one type of aberration is present and accounts for the entire PSF width. While this approach may be useful to qualitatively indicate the criticality of different types aberration, it does not appear representative for the Sentinel-5 instrument (as implied by the manuscript title). The strong dependence on the shape of the PSF, which seems to govern the wide range of ISRF errors, raises the question if the results can be used to predict the ISRF performance for a (more realistic) combination of the investigated aberrations. Do the ISRF-shape errors for spherical and comatic errors (reported in tables 1-3) add up linearly or RSS in systems with both aberrations? The authors shall... 1a) perform analysis with the estimated aberrations of the Sentinel-5 instrument (NIR and SWIR bands), or 1b) justify more clearly why such estimate of Sentinel-5 (in the NIR band) cannot be made. 2) in case of 1a) perform various analysis with cases of mixed aberration to explore the how different aberration components add up 3) if possible, extend the analysis to other types of aberration

Response: The aberrations present in the Sentinel-5/UVNS will vary with wavelength and ACT field point. Further, they will not be characterized experimentally but only in terms of RMS spot size. Although not a realistic case, it was our intention to investigate how different PSF, based on a pure single aberration component, impact the ISRF stability. Therefore, the analysis was intended to be representative in RMS spot size, but not in terms of the exact RMS wavefront error. In the revised manuscript, we will change the analysis to additional Zernike terms and also include mixtures of expected aberrations in Sentinel-5/UVNS and investigate how the different aberration components add up to the overall ISRF knowledge.

Changes in manuscript: We extended the analysis to several other Zernike Polynomials including two mixtures consisting of three individual Zernikes. The motivation and explanation for the Zernike analysis is given in line 298-306. The results of the analysis are given in Table (1) and (2).

**R2: 4. Lack of exploitation of the obtained results (no comparison with classical slit, no flowdown of the results to Level-2 performance, missing interpretation in terms of implications for design improvements).[...]**

The final results are presented in three tables, in which the three figures-of-merit (shape distortion, FWHM and spectral barycentre position) are listed for three assumed aberration scenarios and four selected scenes. The interpretation is limited to commenting on the values listed therein, reporting the expected higher values for higher contrast, the difference between the three aberration cases, and the large values for the CAL-scenes. The latter are found to be are alarmingly high, but no conclusions are drawn for the instrument under investigation. It is merely stated that the performance is compliant for the "S5-ESA-scene", which is not surprising as it features very low contrast. The reader is left clueless about the real performance for higher contrast scenes, or complex albedo variation which the instrument will be exposed to in flight. In fact, it is not even clear if the use of a SH brings any advantage over using a classical entrance slit. It would be important to clarify if the performance gain by homogenisation of the near-field is completely or only partially lost by the induced far-field non-uniformity. No plots of the distorted ISRFs are provided, which would be instructive for the prediction of the impact on retrieved products. It would be interesting to see, how the different types of aberration affect the ISRF for the various scenes (e.g. extension of the wings or skewing the shape). While the changes are probably too subtle to be seen in the ISRF, it should not be difficult to include difference plots w.r.t. the homogeneous reference for some (extreme) cases. It should even be possible to flow down the obtained ISRF distortions to the Level-2 products. This is not easy in absence of a Level-2 processor (or end-to-end simulator), but could be approximated by using so-called gain vectors, which are part of the Sentinel-5 requirement definition. It would also be insightful to present the radiometric errors arising from the distorted ISRFs, at least for the extreme cases (largest and lowest). For this, the monochromatic reference spectrum would be convolved with the obtained (distorted) ISRFs and the difference plotted as a fraction of the true radiance (homogeneous, aberration-free case). The manuscript also falls short on providing implications of the results, and recommendations for design improvements. Can the pupil non-uniformity from SH be limited by design or is it an unchangeable "fact-of-life"? Does it depend on the number of reflections and can be mitigated by extending of SH along the optical axis? How important is the slit width and the focal lengths of the telescope and collimator ? Can the results be used to guide the optical design of the collimator and imager optics, e.g. regarding the types of aberration ?

Response: We agree, the interpretation and conclusions of the result are very limited. In the revised manuscript, we will add a more thorough discussion. We will compare our results to the case with a classical slit and highlight the improvement of the scene homogenization even in the presence of spectrograph illumination variations. We will also point out, that the calibration scenes for which we calculate the ISRF merit functions are only representative of on-ground measurements and don't represent real flight scenarios. As it should be the primary goal of this study to compare with real flight scenarios, we would only present a single Calibration Scene (50%) and emphasize, that the values obtained are without accounting for motion smear and therefore yield exaggerated ISRF distortions.

We will include plots of the distorted ISRF for different types of aberrations and also compare to the case of using a classical slit.

The flow down of the obtained ISRF distortions to Level-2 products is definitely important. We are currently working on another study dedicated to assess the improvement of Level-2 products by employing a slit homogenizer. Therefore, we will publish this topic in a subsequent publication. We will also add a discussion about design recommendations.

Changes in manuscript: We added the "Results and discussion" section (Sect. 5) (lines 309-365).

R2: 5. Invalid conclusions about other missions, which use different SH technology, and based on inadequate comparison of mission requirements.[...]

Two main conclusions drawn by the authors are poorly justified: Conclusion 1): Quote: "A representative scene of the Sentinel-5/UVNS instrument has a rather weak contrast and therefore the instrument fulfils the ISRF specifications in order to meet the Level-2 performance requirements of the mission. In contrast to this, future missions like CO2M have to be compliant with higher contrast scenes with almost a sharp transition from dark to bright slit illuminations." Both parts of the above conclusion... a) There is no criticality for Sentinel-5 as it will only see low contrast b) CO2M will experience large errors, because it will see much larger contrast ...cannot be justified by the presented analysis: a) The low-contrast scene for Sentinel-5 (no source and details given) is likely from the system requirement document (please confirm). Such reference scenes are often specified to constrain individual error sources (e.g. straylight), but do not necessarily represent realistic geophysical scenarios. It appears certain that a Sentinel-5 measurement in the SWIR band near water bodies (coast lines or lakes), with an instantaneous field-view of 2.5 km smeared over 7.5 km will yield much larger effective contrast than the scene referred to as "S5-ESA-scene". b) CO2M is not a "high-contrast mission" as opposed to Sentinel-5. All nadir-looking pushbroom spectrometers look at the same Earth scenes, and the effective scene contrast observed over the integration time depends on the ratio of the slit projection and the ALT sampling distance. This ratio is comparable for bot, S5 and CO2M, and therefore the "smearing" of the contrast will be similar (not a sharp transition as claimed). The formulation of requirements by means of contrast scenes is often driven by straylight requirements, which may be more stringent for CO2M (due to deeper absorption structures in the SWIR-2 around 2.0 µm). However, such contrast scenes cannot be regarded as "representative Earth scenes". In case of CO2M the specified scenes exhibit an extreme albedo contrast (factor of 8) that is only observed over coast lines (and then mitigated by motion smear). The authors should refrain from performance prediction based on the interpretation of requirement documents, especially for missions out of the scope of this investigation (see below). Conclusion 2): Quote: "The application of the slit homogenizer

for missions with high contrast scenes (CO2M) will impose strong variations in the spectrograph pupil and will result in large errors in the ISRF and hence significantly degrades the accuracy in the retrieval of the atmospheric composition and therefore the mission product." This speculative statement is most likely false for the following reasons (on top of the ones given above): a) The presented model for ISRF distortions is based on waveguide propagation along a mirror-based SH. Such a device is not foreseen for CO2M, where a fibre based slit will be employed instead. The model developed in this paper is not valid for light propagation in multimode fibres. In fact, measurements of transfer functions with such a fibre based, two dimensional slit homogeniser (2DSH) have not shown interference patterns as shown in Fig. 3b. (see S.Amann et al., Characterization of fiber-based slit homogenizer devices in the NIR and SWIR, Proceedings Volume 11180, International Conference on Space Optics -ICSO 2018; 111806C (2019) https://doi.org/10.1117/12.2536147 b) Scene-dependent far-field effects from scene non-uniformity are also expected for fibre slits, but are typically less pronounced and can be mitigated by adjusting the fibre length (see G. Avila, "FRD and scrambling properties of recent non-circular fibres," Proc. SPIE 8446, Ground-based and Airborne Instrumentation for Astronomy IV, 84469L (24 September 2012); doi: 10.1117/12.927447). They are related to the fibre modes and are expected to show lower frequency variations than the ones found in this study. They can also be mitigated by fibre bending. It is understood that the authors seek to underline the importance of their results by pointing out the relevance to other missions. While this is legitimate, CO2M is not an appropriate mission for comparison. I am aware of only one other mission considering the implementation of a mirror-based SH: The Geostationary Carbon Cycle Observatory (GeoCarb), which is not quoted in the manuscript. Its step-and stare slit-scan strategy is likely to be more critical regarding the discussed effects than Sentinel-5, because of the absence of motion smear. Unless the authors can justify the validity of their propagation model for rectangular multimode fibres, it is suggested to remove speculative statements about CO2M's in-orbit ISRF stability performance. Instead it is proposed to make reference to GeoCarb (and make the team aware of a potential error source not yet considered), e.g.: - B.Moore, "The GeoCarb Mission," in 14th International Workshop on Greenhouse Gas Measurements from Space, (2018) - J. Nivitanont et al: Characterizing the Effects of Inhomogeneous Scene Illumination on the Retrieval of Greenhouse Gases from a Geostationary Platform. Poster presented at the 4th International Workshop on Greenhouse Gas Measurements from Space, (2018) Finally, I propose to add a conclusion that is currently missing: Both, the transfer function shown in Fig. 3b, as well as the pupil intensity distributions in Fig. 6 should be accessible to measurement employing an appropriate test bench. It is assumed that the two SH devices in Sentinel-5 are now mature enough to be tested (btw. please note the existence of two different such slits in the manuscript). Far-field measurements with these devices could be used to verify the derived model, and to quantify the ISRF errors expected from measured pupil intensity variation. If the authors agree, I suggest to include such proposal and give indications on how to implement an appropriate measurement.

Response: We agree and remove the conclusion and connections that we tried to establish to the CO2M mission. Our intention was to point out the limitations of the 1D-Slit Homogenizer in terms of scrambling performance over extreme albedo contrasts. However, as mentioned by the referee, CO2M is not employing a mirror based slit homogenizer but a fibre-based 2DSH and our model is not applicable to such devices.

We follow the referee's recommendation and include a suggestion on how to experimentally verify the expected pupil intensity variations.

Changes in manuscript: The references to the CO2M mission have been entirely removed from

the manuscript. We briefly mention the work on another scene homogenization techniques based on multimode fibres (lines 362-365.). The experimental suggestion for a dedicated measurement setup to experimentally verify our simulation results is given in lines 355-361.

**2.2 Technical corrections / Editorial Comments**

R2: 1. p. 1; "The spectral accuracy" is not well defined so far.

Response: We will change the sentence.

Changes in manuscript: see lines 1-3.

R2: 2. 1. 3-5: "As the ISRF is the direct link between the forward radiative transfer model" >> add: ", used to retrieve the atmospheric state..."

Response: Done.

Changes in manuscript: see lines 3-4.

R2: 3. l. 14: "By homogenizing the slit illumination, the SH moreover strongly modifies the spectrograph pupil as a function of the input scene" >> insert "illumination" after "pupil"

Response: Done.

Changes in manuscript: see line 14.

R2: 4. l. 16: "type" >> "type"

Response: Probably it is meant to change it to "types" which will be done.

Changes in manuscript: see line 16.

R2: 5. l. 19 "As in most space based imaging spectrometer" >> is too general, e.g. imaging FTS (e.g. IASI) are not affected (no slit) Also indicate the difference to scanning spectrometers, like SCIAMACHY

Response: We will point out the difference to imaging FTS and scanning spectrometers.

Changes in manuscript: see lines 20-23.

R2: 6. 20: "spectrometer" >> "spectrometers"

Response: Obsolete due to new formulation above (6.)

R2: 7. l. 20: "gets imaged" >> "is imaged" (is the purpose)

Response: Done.

Changes in manuscript: see line 20.

R2: 8. l. 21: delete "eventually"

Response: Done.

R2: 9. l. 22: "gets convoluted" >> "is convolved"

Response: Obsolete due to new formulation below (10.).

R2: 10. l. 22: better: "The limited spectral resolving power of the instrument arising from diffraction and aberration is describe by a convolution of the slit image with the spectrometer and detector point spread functions (PDF).

Response: Done.

Changes in manuscript: see line 23-25.

R2: 11. l. 24: "The resulting intensity pattern on the FPA in the spectral direction is called the instrument spectral response function (ISRF)." -> This is not a universal definition. According to ESA definition, this is the ISMF (Instrument Spectral Measured Response), which is not measurable continuously, but sampled by the detector pixels. ESA defined ISRF for each detector pixel as a continuous function of wavelength, defined as the individual pixel's response at a given wavelength. In absence of aberrations, this ISRF is a mirror of the ISMF (inverted on the spectral scale), but in reality this is not the case. For definitions please refer to : Caron, J., Sierk, B., Bézy, J.L., Löscher, A., and Meijer, Y., The CarbonSat candidate mission: Radiometric and spectral performances over spatially heterogeneous scenes, Proceedings of the International Conference on Space Optics (ICSO), Tenerife, Spain, 2014 It is also suggested that the authors read and (if found appropriate) cite the following reference, which highlights the issue of inhomogeneous slit illumination for a relevant airborne instrument: Gerilowski, K., Tretner, A., Krings, T., Buchwitz, M., Bertagnolio, P. P., Belemezov, F., Erzinger, J., Burrows, J. P., and Bovensmann, H.: MAMAP – a new spectrometer system for column-averaged methane and carbon dioxide observations from aircraft: instrument description and performance analysis, Atmos. Meas. Tech., 4, 215–243, https://doi.org/10.5194

Response: We will soften the formulation and indicate, that in our the model, the ISRF is defined as the slit intensity pattern on the FPA in the spectral direction. We will add the reference to the proposed paper and indicate, that our definition is only valid in the absence of smile effects. We will also cite the paper of Gerilowksi et. al in the discussion of non-uniform scenes in ACT direction.

Changes in manuscript: For the definition of the ISRF, see lines 25-28. The citation of Gerilowski et al. in the context of non-uniform scenes in ACT is given in lines 67-69 and 230-231.

R2: 12. l. 28: "Figure 2 depicts a representative Top- of-Atmosphere spectrum" >> Be more specific, indicate the albedo and SZA.

Response: Done.

Changes in manuscript: see line 33.

R2: 13. l. 28: "in the SWIR wavelength band" -> "for the Sentinel-5 SWIR-3 spectrometer, used for retrieval of CH4 and CO" >> The plotted spectral band is one out of 2 SWIR bands of the Sentinel-5 mission, and other missions have yet different

**band definitions.**

Response: Done.

Changes in manuscript: see line 34.

R2: 14. l. 29: "entering a space-borne instrument" -> "incident on a space-borne instrument's entrance aperture" "high-resolution" -> "monochromatic" or "unconvolved"

Response: Done.

Changes in manuscript: see line 34-35.

R2: 15. l. 30: "for every monochromatic stimulus" Not clear what is meant by this. It seems to refer to the incident spectrum as a continuum of monochromatic stimuli. Better replace by "for any given wavelength"

Response: Done.

Changes in manuscript: see line 36.

R2: 16. l. 31 "Whenever the ISRF shape deviates from the on-ground characterized shape..." while the ISRF has been introduced as a mathematical convolution kernel, on-ground characterisation is mentioned "by the way" in a side sentence. It should be mentioned in the text that the ISRF is a wavelength- and field-of-view dependent instrument characteristic (varying with wavelength and ACT field position), and is determined by onground characterisation prior to launch.

Response: We will add the FPA position dependent ISRF characteristics and the on-ground characterization prior to launch.

Changes in manuscript: see lines 36-39.

R2: 17. l. 32 "..., which serves as a basis for the applied retrieval algorithms." -> "...from which the Level-2 products are retrieved". (measurements are the input to, not the basis of an algorithm). Since the paper mainly addresses the Sentinel-5 mission and the SWIR bands, the retrieved Level-2 products shall be mentioned (CH4, CO columns).

Response: Done.

Changes in manuscript: see line 40.

R2: 18. l. 33: "The along track motion of the satellite accounting for the spectral direction of the spectrometer serves as an averaging and smearing effect of the scene" "The along track motion of the satellite during the integration times results in a temporal averaging of the ISRF variation, which reduces the impact of scene heterogeneity." Also mention here that the impact of albedo variations depends on the ratio of the instantaneous field-of-view (IFOV) and the sampling distance in ALT. Please indicate these numbers for Senyinel-5 (FoV=2.5km, ALT SSD = 7.5km).

Response: Done.

Changes in manuscript: see lines 41-44.

R2: 19. l. 36: "are less vulnerable to contrast in the Earth scene" -> Indicate the reason: The effect depends on the ratio of spatial sampling distance (in this sentence "scan area"), and the instantaneous field of view (see comment above)

Response: We will add this information.

Changes in manuscript: see lines 45-46.

R2: 20. l. 36: "In contrary," - > in contrast

Response: Done.

Changes in manuscript: see line 46.

R2: 21. l. 37: "...define a set of stringent requirements on the inflight knowledge and stability of the ISRF." >> add the reasone before: "...high resolution hyperspectral imaging spectrometers with IFOV comparable to the sampling distance (or scan area) are more strongly affected and therefore define..."

Response: Done.

Changes in manuscript: see lines 46-47.

R2: 22. l. 38: "will introduce biases in the Level-2 data" add "and pseudo-random noise" after "biases", which is actually the main impact over an ensemble of measurements.

Response: Done.

Changes in manuscript: see lines 48-50.

R2: 23. l. 39: "For the 2017 launched Sentinel-5 Precursor (S5P) satellite..." ->"For Sentinel-5 Precursor (S5P) satellite, launched in 2017,", and add reference for mission description, e.g. J. P. Veefkindet al., TROPOMI on the ESA Sentinel-5 Precursor: A GMES mission for global observations of the atmospheric composition for climate, air quality and ozone layer applications, Remote Sensing of Environment, Volume 120, p. 70-83 (2012)

Response: Done.

Changes in manuscript: see line 50-51.

R2: 24. p. 3; Figure 2: >> Suggested to plot the monochromatic TOA spectral radiance and the convolved, simulated measurements should be plotted here as well. >> Briefly explain the origin of the spectral structure, indicating the absorption features by CH4, CO, and H2O.

>> It would be instructive to include an over-plot of a spectrum with distorted ISRF for a realistic scene contrast, and to include a difference plot w.r.t. the homogeneous case

Response: We add the explanation of the spectral structures.

The considerations about radiometric errors due to ISRF distortions in the context of non-uniform

scenes will be presented in a subsequent study. For this manuscript we will focus on the ISRF knowledge in terms of the figures of merit as defined in the manuscript and hence will not go into further detail about the variation of the spectra for different ISRFs.

Changes in manuscript: see caption of Figure 2.

R2: 25.1. 43: "Noel et al. (2012) quantify the retrieval error for the Sentinel-4 UVN imaging spectrometer for the tropospheric O3, NO2, SO2 and HCHO." >> Indicate that Sentinel-4 is not yet launched (adding "upcoming") and that these results are based on simulations, not real measurements (in contrast to the TropOMI results quoted before). Also introduce the not yet defined acronym "UVN".

Response: Done. The acronym UVN will be added, where mentioning Sentinel-5/UVNS for the first time.

Changes in manuscript: see line 34.

R2: 26. 1 45: "They propose a software correction algorithm which is based on a wavelength calibration scheme individually to all Earthshine radiance spectra" - add comma in -> "algorithm, which..." - add "applied" and replace "Earthshine" by "Earth" (even though used in the reference)-> "...individually applied to all Earth radiance spectra..."

Response: Done. Changes in manuscript:see lines 57-58.

R2: 27. l. 49,50: "...to mitigate the effect of non-uniform scenes." add "in along-track direction", as the S5's SH only homogenises in ALT. It shall be mentioned, that non-uniform scenes in ACT direction also result in ISRF distortion in presence of smile distortion in the image plane. This is e.g. explained in the already quoted Caron et al. 2014 (see reference above).

Response: We will mention the impact of heterogeneous scenes in ACT direction and refer the reader to Gerilowski et. al (2011) and Caron et al. (2017).

Changes in manuscript: see lines 67-69 and 230-231.

**R2: 28. l. 51: move "in the along track direction (ALT) of the satellite flight motion" after "Earth radiance"**

Response: Done.

Changes in manuscript: see line 63.

R2: 29. l. 53: "...mirrors is of  $b = 240 \ \mu m$  (NIR)," -> replace "of" by "has dimensions of" Proposed: "The two parallel rectangular mirrors composing the entrance slit have a distance of  $b = 240 \ \mu m$  (NIR), side lengths of 65 mm in ACT and a length of 9.6 mm along the optical axis".

Response: Done for the values in the SWIR-3 channel.

Changes in manuscript: see line 65.

R2: 30. l. 54: "gets scrambled" -> "is scrambled by multiple reflections"

Response: Done.

Changes in manuscript: see line 66.

R2: 31. l. 55: "For a realistic reference Earth scene of the Sentinel-5/UVNS mission" >> Please indicate the specifics of the scene (albedo image or artificial contrast)?

Response: We will revise the whole section of the description on the applied scene cases and refer to the answer in General Comment (2).

Changes in manuscript: See section (3), lines 214-230.

R2: 32. l. 56: "the total in orbit ISRF shape error budget is < 2 %, the relative Full width half Maximum (FWHM) error < 1 % and the centroid error in the NIR 0.02 nm" >> Although hidden in the word "budget" state more clearly that these are requirements, not resulting performances.

Response: Done.

Changes in manuscript: see line 70.

R2: 33. l. 61: "We present an end-to-end model of the Sentinel-5/UVNS NIR channel (760 nm)." Please justify why the model (resp. its application) is restricted to the NIR band. Also replace, or add equivalent plots of the NIR band, as for SWIR in Fig. 2.

Response: As we will change the analysis to the SWIR-3 channel, we assume that this comment becomes obsolete.

Changes in manuscript: The discussion on why the SWIR-3 is the most critical one is given in lines 342-344.

R2: 34. l. 63: "consequently implies a scene dependency in the optical PSF" "implies" -> "results in"

Response: Done.

Changes in manuscript: see line 78.

R2: 35. l. 65: "spectrograph pupil intensity distribution" -> "intensity distribution across the spectrograph pupil"

Response: Done.

Changes in manuscript: see line 79-80.

**R2: 36. l. 76: "contains details" -> "describes"**

Response: Done.

Changes in manuscript: see line 91.

R2: 37. l. 78: "The second part focusses on the novel modelling technique of the spectrograph optics." Not understood. Is "novel modelling technique" referring to the previous sentences, or is it announcing a new technique to be established in the paper. In the latter case, better write: "In the second part a novel modelling technique of the spectrograph optics is introduced".

Response: Here we wanted to refer to the technique established in the paper. Therefore we will use the recommendation as proposed by the Referee.

Changes in manuscript: see line 93-94.

R2: 38. l. 87: Please mention that equation 2 follows from equation 1 with the simplifying assumption of a square entrance pupil. This is currently hidden in a side remark on line 92. Clarify the calculated quantity  $\tilde{U}_f$ ,  $\theta$  (electrical field?), currently referred to as "the Airy disk"

>> Please clarify if this propagation model has been verified against measurements of the SH transfer function.

Caption of Fig. 3 "...highly dependent on interference effects" -> "are strongly affected by interference effects, resulting in a complex illumination pattern at the slit exit". You should mention that this is not a new finding, but a known characteristic of such SH device, and that the interference pattern, although not uniform, already represents an improvement over no scrambling in a classical slit.

Response: We will add a sentence and mention that the airy disc is the field distribution in the slit plane and is given by the Fourier transform of the electric field over the entrance pupil. As the telescope pupil is actually of rectangular shape, we will change the word airy pattern to diffraction pattern.

We will mention, that a full end to end verification of the propagation is still missing. However, an initial approach by ITO Stuttgart to validate the performance model in a breadboard activity gave confidence on the approach of the optical model (see Irizar et al. (2019)).

Changes in manuscript: The explanation of the airy disc is given in lines 101-102. Note that we changed the formulation from airy disc to diffraction pattern.

The reference to the measurements of ITO Stuttgart Published in Irizar et al. ar given in lines 123-124.

We provide a thorough comparison to a classical slit in the Section (5) and also discuss the limitation of the SH based on the remaining fluctuations of the SH transfer function.

R2: 39. l. 121: ". Independent of the applied scene in ACT, the telescope pupil is retrieved again at the spectrograph pupil despite a magnification factor and a truncation of the electric field at the SH entrance plane, which leads to a slight broadening and small intensity variations with a high frequency in angular space (Berlich and Harnisch, 2017)." >> split in 2 sentences

Figure 4.: Indicate in the caption the astigmatism of the collimator, which is adjusted to the slit length.

Response: Done.

Changes in manuscript: see caption of Figure 4.

R2: 40. l. 134: "...spectrograph pupil will be altered with respect to the telescope pupil" The manuscript frequently refers to "altering" telescope and spectrograph pupils, although these are optical-geometric terms that do not depend on the illumination. For better correctness it should be rephrased in terms of "pupil illumination"

Response: We corrected the respective formulations throughout the manuscript.

R2: 41. l. 134: "A general case for the connection between slit exit plane and spectrograph pupil plane is considered by Goodman (2005, p. 104)" The coordinates in Eq. 5 seems to be for the slit exit and spectrograph pupil, respectively, which differs from Eq. 2-3, where x,y, denote coordinates in the telescope pupil and u,v those in the slit exit. Please clarify the coordinates or use indices to indicate the difference.

Response: We agree, that our notation is confusing. We changed our coordinate notation in the following way:  $x_t, y_t$  are the coordinates at the telescope pupil,  $u_a, v_a$  at the SH entrance plane,  $u_b, v_b$  at the SH exit plane and  $x_s, y_s$  at the spectrometer pupil plane. Note, that we also observed a typo in equations (5,6,10), which was the incorrect factor  $(u^2 + v^2)$  in the integral.

Changes in manuscript: We changed the coordinate formulation throughout the manuscript. Note that we also observed small typos in equation (4),(5),(10),(14) and (17) which are showed in the changelog of the manuscript at the end of the document.

R2: 42. l. 139: "where k is the wavevector of the incoming wave,  $\lambda$  is the wavelength and f is the focal length of the lens" Quantities already defined above.

Response: We erased the sentence.

**R2: 43. l. 144, Eq. 6: Is it necessary or convenient to keep 2pi and lambda in the argument of the field distribution ?**

Response: It was our intention to highlight the coordinate transformation that was necessary in order to solve the equation. In the revised version we define:  $x'_s = \frac{2\pi}{\lambda f} x_s$  and  $y'_s = \frac{2\pi}{\lambda f} y_s$ .

Changes in manuscript: We introduce the coordinate transformation in line 188.

R2: 44. l. 145: By ending the section with an uncommented equation, the reader is left with the question what is the conclusion so far (or the purpose of the calculation). It should be noted that this is an intermediate result, which will be further propagated and refined in the following (for collimator astigmatism).

Response: We will add a comment on the derived formulation.

Changes in manuscript: see lines 168-169.

**R2: 45. l. 162: "straight forward" -> "straightforward"**

Response: Done.

Changes in manuscript: see line 182.

R2: 46. l. 175: "Further, we model the dispersive element as a 1D binary phase diffraction grating." >> Indicate if this choice is relevant to the actual Sentinel-5 instrument. An image of the binary grating structure would be useful here (also for explaining the quantities used in the text).

Response: We will add an explanation, that the used model for the grating is a simplified assumption. We will add the real grating configurations of the Sentinel-5/UVNS in the SWIR-3.

Changes in manuscript: see line 209-211.

R2: 47. l. 191: "...representative Earth scene for the Sentinel-5/UVNS instrument provided by ESA (S5-ESA-scene)..." The "scenes" referred to here need to be further described. Is it the artificial contrast scene (bright and dark reference spectrum)? In that case they should not be called "representative Earth scene", but a step transition scene with contrast factr representative for the mission requirements.

Response: We will revise the whole section and refer to the answer in General Comment (2).

Changes in manuscript: see Sect. 3.

R2: 48. l. 194: "Fig. 5 shows the top of atmosphere (ToA) radiance level given by a realistic Earth scene for the Sentinel-5/UVNS instrument." This is likely showing the contrast for one wavelength (supposedly spectral continuum level), which can vary significantly across the spectrum (zero in case of saturation). Please clarify in the text.

Response: We will revise the section and refer to the answer in General Comment (2).

Changes in manuscript: see Sect. 3.

R2: 49. l. 195: "Due to smearing of the satellite's movement, this scene has a significantly lower contrast than the calibration scenes"

>> Indicate this by plotting the convolution of the contrast with a boxcar function of the motion smear, which would show the contrast the instrument sees during integration time.

Response: This will be done in the revised section (see General Comment (2)).

Changes in manuscript: See Figure 5.

R2: 50. l. 196-200: The remark about the CO2M seems misplaced here, and should be moved to the discussion of the results (if maintained at all). It seems incorrect to equate a "calibration scene" with stationary contrast in the slit with a "representative scene of another instrument (especially with a different type of SH, see below). While it is true that nonhomogeneous scenes are more critical for CO2M, there will also be smoothing by morion smear, and a sharp transition cannot be observed. This is different from step-and-stare instruments (e.g. GeoCarb), which could be mentioned here.

Fig. 5: It is still unclear, how the relatively flat "S5-ESA-scene" was derivedn(origin and processing, e.g. convolution with motion smear, assumption of slit projection, etc.). Please clarify.

Response: We removed the reference to CO2M (see General Comment (5)). The derivation of the scene will be explained in more detail in the revised version of the manuscript (see answer to General comment(2)).

Changes in manuscript: For the derivation of the scene, see Sect. 3.

R2: 51. l. 201: "Figure 6 depicts the pupil intensity distribution in the NIR (760 nm) for the applied test scenes" >> Indicate that these are simulations based on the equations derived before.

Response: We will indicate that the results are based on simulation.

Changes in manuscript: See caption of Figure 6 and line 236.

R2: 52. l. 202: remove "is happening" -> "due to the absence of interaction, i.e. reflection, with the SH"

Response: Done.

Changes in manuscript: see line 238.

R2: 53. l. 204: "retrieved" -> "preserved"; "Contrary" -> "In contrast"

Response: Done.

Changes in manuscript: see line 239.

R2: 54. l. 215: "We know from ray tracing simulation predictions the PSF size on the FPA of the Sentinel-5/UVNS NIR channel, which in a simplified model is given by the standard deviation of a normal distribution. As the actual aberrations present in the system are yet unknown,..." - > It is hard to believe that the aberrations for the Sentinel-5 instrument are completely unknown at this point (so far into the project). The PSF usually depends on field position (and wavelength), and should be well characterised by the optical analysis. It is understood that the use of Gaussians is convenient for the mathematical analysis, but it would be good to verify the results are robust against more representative PSF.

Response: See General Comment (3).

Changes in manuscript: As mentioned above in General Comment (3).

Eq. 14: - Explain that (u,v) are now the coordinates at the focal plane - The intensity  $I_{theta}$  is the square of the absolute value of  $U_{FPA}$ , which has no dependence on theta in Eq. 14. Please clarify why it appears as a function of theta in Eq. 15. It might be good to write here the one-dimensional equation for  $I(\theta, \nu)$ , as it represents the final result for the ISRF distortion.

It is not clear how the aberrations to demonstrate the impact of inhomogeneous pupil illumination were selected (randomly, analysis or for convenience)? It should be possible to make realistic estimates on the aberrations present in the Sentinel-5 instrument. This would enhance the credibility of the results regarding ISRF impact.

Response: In order to avoid confusion we change (u,v) to (s,t) and mention that they are the coordinates at the FPA plane.

In the derivation of the field distribution of the SH exit plane (Eq. 4) we indicated, that the calculation is made for a single incidence angle  $\theta$  onto the SH entrance plane. As the referee rightly mentions, this reference was not made in the subsequent steps, which still describe the propagation of a single SH entrance incidence angle  $\theta$ . It will be added to the equations describing the propagation through the spectrograph.

Changes in manuscript: See new formulation of the coordinates in line 267.

For better reading, we erased equation (6) of the previous version of the manuscript and mention it directly in equation (9) of the revised manuscript. The angle  $\theta$  was added accordingly to equation (9,11,12,14).

R2: 55. l. 262: "result" -> "results"

Response: Obsolete due to new formulation.

Changes in manuscript: see lines 291-295.

R2: 56. l. 264: << "As a direct comparison of the difference between an ISRF calculated with a PSF disturbed by aberrations and a PSF given as a pure normal distribution is problematic, we rather compare the errors relative to a homogeneous scene." >> Please explain in more detail why it is "problematic". It was stated above that assuming a Gaussian PSF would "neglect the non uniformity in the pupil and the spectrometer aberrations."

Table 1-3: - It is not clear why the errors are so large for the Gaussian PSF case. If the ISRF distortion originates from scene-dependent weighting aberrations, then this case should not yield large errors. - Please indicate in the text how these results compare with the requirement for ISRF knowledge. - Please plot the distorted ISRFs corresponding to the results in the table (at least the extreme ones) - It is suggested to also include plots showing radiometric errors resulting from such distortions

>> Does the result, which predicts large ISRF knowledge errors from the "calibration scenes", have any implications on the on-ground calibration of the instrument? Please elaborate.

Response: A direct comparison of ISRF containing different kind of aberrations creates a systemic error as the ISRFs are based on different PSF (due to the different aberrations) and are not comparable even in the absence of non-uniform scenes. However, the ISRF will be extensively characterised on-ground for homogeneous scenes and hence aberrations are compensated by calibration. We want to investigate how the ISRFs based on several Zernike terms behave under the condition on non-uniform scenes and how the ISRF errors evolve with respect to each specific homogeneous ISRF. Therefore we calculate the relative change in ISRF figure of merit functions and not directly compare differently aberrated ISRFs with each other.

The errors presented in the tables represent the ISRF errors combining the effect of remaining SH exit non-uniformity (near-field) and non uniform spectrograph illumination (far field) in combination with optical aberrations. Therefore, the case of a constant gaussian PSF contains only the errors of the remaining near-field non-uniformity, whereas the case with Zernike aberrations and propagation through the spectrograph contains both, near field and far field errors. The relative difference in the ISRF stability between these two cases gives an estimation on the resulting far field errors contributions which is the main part of this study.

The previous version of the manuscript showed ISRF distortion values for calibration scenes that are used for the experimental validation of the SH performance model and don't represent real flight scenarios (see General Comment(2)). Therefore, the results are useful in the upcoming characterization and verification campaign to detect and understand possible discrepancies in the SH model and experimentally measured results for non-uniform scenes of such kind.

We will include plots of the distorted ISRFs.

As said in the response to General Comment (4), the investigations wrt the impact on Level-2 are ongoing and are planned to be published in a dedicated paper.

Changes in manuscript: We emphasize more clearly, that the errors shown in Table (1) and (2) are the combined errors of the near-field and far-field variations. See caption of Table (1) and line 296-297.

The plots of the distorted ISRF are given in Figure 7.

The application of the calibration scene is mentioned in lines 233-235 and 311-312.

R2: 57. l. 277: "gets significantly higher" -> "becomes significantly higher"

Response: As we revise this section, this point becomes obsolete.

R2: 58. l. 279: "...comes only by..." - > "...comes only from..."

Response: Obsolete as we revised the section.

R2: 59. l. 285: "A scene dependency of the spectrograph pupil will lead to similar ISRF distortion as due to non-uniform slit illuminations" >> This could, but was not shown here. The authors should provide calculations for ISRF distortions for a classical slit, in order to compare and support this claim.

Response: We will compare the results with the case of a classical slit.

Changes in manuscript: The comparison with a classical slit is given in Table (1) and (2) as well as in Fig. 7a.

R2: 60. l. 281: "We also conclude, that for the Sentinel-5/UVNS instrument the impact of this effect is of second-order and doesn't degrade the performance of the SH significantly." This conclusion should be restricted to the reference scene used, not the Sentinel-5/UVNS instrument. Independent on the requirement formulation, the instrument might be exposed to larger contrast than used in this study.

Response: See General Comment (2).

Changes in manuscript: See Sect. 3. We will mention, that larger contrasts scenes are excluded from the Sentinel-5/UVNS mission requirements in lines 336-338 and 372-373.

R2: 61. l. 328: Duplication in reference: Goodman, J. W.: Introduction to Fourier optics, Introduction to Fourier optics, 3rd ed., by JW Goodman. Englewood, CO: Roberts & Co. Publishers, 2005, 1, 2005.

Response: Done.

Changes in manuscript: see line 425.

**References**

Irizar, J., Melf, M., Bartsch, P., Koehler, J., Weiss, S., Greinacher, R., Erdmann, M., Kirschner, V., Albinana, A. P., and Martin, D.: Sentinel-5/UVNS, in: International Conference on Space Optics — ICSO 2018, edited by Sodnik, Z., Karafolas, N., and Cugny, B., vol. 11180, pp. 41 – 58, International Society for Optics and Photonics, SPIE, https://doi.org/10.1117/12.2535923, URL https://doi.org/10.1117/12.2535923, 2019.

**Slit homogenizer introduced performance gain analysis based on Sentinel-5/UVNS spectrometer**

Timon Hummel1,2, Christian Meister2, Jasper Krauser2, and Mark Wenig1 1Meterological Institute LMU Munich, Theresienstraße 37, Munich, Germany 2Airbus 
[revised manuscript text omitted]

---

## Author Response (AR2)

**Author response to associate editor**

July 8, 2021

We thank the associate editor for the comments and suggestions. We answered all points addressed in the review and implemented your suggestions. All page and line numbers refer to the tracked changes file of the revised manuscript. Appended to this document is a differential view between the revised version after the referee review and the revised version after the editor review for convenient tracking of the applied changes.

**1) Introduction (page 1, line 23)**
**The expression 'slit based spectrometers' is rather unfamiliar, perhaps it should be changed to something like: '... spectrometers incorporating an entrance slit as optical element ...'.**

Response: Instead of "slit based imaging spectrometer" we rather call them "grating based spectrometers".

Changes in manuscript: In grating based imaging spectrometers, the Earth ground scene is imaged by the telescope onto the instrument entrance slit plane.

**2) Introduction (page 1, lines 24 following):**
**Is the text '... scanning mirror or a push-broom configuration, where different areas of the surface are imaged as the satellite flies:forward ...' actually describing the situation correctly? Is not rather one dimension (along track) of the scanning always provided by the motion of the satellite and the other (cross track) either by scanning or imaging?**

Response: We think, that both expressions are correct and do not contradict one another. As suggested by referee # 2, our intention was to briefly mention the scanning mechanism without going into too much detail.

**3) Introduction (page 2, line 42 ):**
**What is a monochromatic spectrum?**

Response: A Monochromatic spectrum denotes the top of atmosphere spectrum before convolving with the ISRF. The term was suggested at this point by referee 2.

**4) Introduction (page 2, line 56 ):**
**Explain 'IFOV'**

Response: The abbreviation of IFOV will be added when mentioning the instantaneous field-of-view in line 52.

Changes in manuscript: The impact of e.g. albedo variations depends on the instantaneous field-of-view (IFOV) and the sampling distance in ALT (for Sentinel-5/UVNS: FoV = 2.5 km, ALT

SSD = 7 km).

**5) Sect. 2.2, (page 9, line 173 ):**
**'... are based ...'**

Response: Done

Changes in manuscript: In contrast, the variations in the spectrograph illumination are based on a geometrical reallocation of the angular distribution of the light exiting the SH in combination with interference effects in the spectrograph pupil plane.

**6) Sect 2.4, (page 11, lines 236, 237):**
**'... the simplified approach is valid also for this case (i.e. the immersed grating) ...'?**

Response: We will add the explanation, that this is also the case for the immersed grating of Sentinel-5/UVNS.

Changes in manuscript: The simplified approach is also valid for this case, as the SH does not affect the general behaviour of the grating.

**7) Sect. 4 (page 17, line 348 following):**
**The sentence starting 'Therefore, although not a realistic case ...' does not appear to be complete.**

Response: Done.

Changes in manuscript: Therefore, although it is not a realistic case, we impinge pure aberrations of a single type in order to determine critical Zernike terms for the ISRF stability.

**8) Page 22, Table 2:**
**The table is lacking a proper and descriptive caption.**

Response: We will repeat parts of the description in table 1 and mention the different scenes and the information, that the values are exaggerated with respect to real flight scenarios.

Changes in manuscript: 50 % CAL scene - ISRF stability. The presented errors combine the remaining SH exit non-uniformity (near-field) and effects due to the variations of the spectrograph pupil illumination (far-field). The strength of the aberrations are chosen such that the spot size matches the case of a PSF size of 6.85 µm (80 % EE). Remark: ISRF values are exaggerated with respect to real flight scenarios. Calibration scenes are used for on-ground SH performance validation.

Additional authors changes:

1) Note, that we changed the name of a variable in eq. (17,18,19) from $y$ to $\lambda$. We want to emphasize that by the introduced transformation, the ISRF is now given in units of wavelength ($\lambda$) as also depicted in figure 7.)

2) We would like to add an additional co-author who contributed to the major revision of the previous referee review. The additional co-author will be added in the revised manuscript and the tracked changes file.